



# Ozone recovery effects on mesospheric dynamics in the southern hemisphere

Ales Kuchar[1], Gunter Stober[2,3], Dimitry Pokhotelov[4], Huixin Liu[8], Han-Li Liu[7], Manfred Ern[9], Damian Murphy[10], Diego Janches[5], Tracy Moffat-Griffin[11], Nicholas Mitchell[11,12], and Christoph Jacobi[6]

[1]Institute of Meteorology and Climatology, BOKU University, Vienna, Austria
[2]Institute of Applied Physics, Microwave Physics, University of Bern, Switzerland
[3]Oeschger Center for Climate Change Research, Microwave Physics, University of Bern, Switzerland
[4]Potsdam Institute for Climate Impact Research, Member of the Leibniz Association, Potsdam, Germany
[5]ITM Physics Laboratory, NASA Goddard Space Flight Center, Greenbelt, MD, USA
[6]Institute for Meteorology, Leipzig University, Leipzig, Germany
[8]Department of Earth and Planetary Science, Kyushu University, Japan
[7]High Altitude Observatory, National Center for Atmospheric Research, Boulder, CO, USA
[9]Institute of Climate and Energy Systems - Stratosphere (ICE-4), Forschungszentrum Jülich, Jülich, Germany
[10]Australian Antarctic Division, Kingston, Tasmania, Australia
[11]British Antarctic Survey, Cambridge, CB3 0ET, UK
[12]Department of Electronic and Electrical Engineering, University of Bath, Bath, UK

**Correspondence:** ales.kuchar@boku.ac.at

**Abstract.** The recovery of Antarctic ozone, primarily driven by the Montreal Protocol, has significantly altered stratospheric circulation. However, its effects on the mesosphere and lower thermosphere (MLT) remain underexplored. Here, we use over a decade of meteor radar observations from Davis (68.6°S, 78.0°E), Rio Grande (53.7°S, 67.7°W), and Rothera (67–68°S, 68°W), and whole-atmosphere model outputs (GAIA and SD-WACCM-X), complemented by satellite and reanalysis datasets, to assess trends in mesospheric winds. Our results reveal a significant delay in the spring transition from westerly to easterly zonal winds at approximately 82 km. This pattern coincides with changes in gravity wave momentum flux and stratospheric winds, suggesting a link between stratospheric ozone recovery and mesospheric dynamics. Comparisons with reanalysis datasets (MERRA2 and ERA5) further validate these findings despite their limited vertical extent and assimilated observations. Model comparisons reveal that, while both GAIA and SD-WACCM-X models have limitations in reproducing the basic climatology of the mesosphere, GAIA shows a somewhat better agreement with the observed trends. These results highlight the continuing effects of ozone recovery on upper atmospheric circulation and the need for improved representation of wave-driven coupling processes in climate models.

## 1 Introduction

The Mesosphere and Lower Thermosphere (MLT) region, spanning altitudes from approximately 70 to 110 kilometers above the Earth's surface, represents a critical interface between the middle atmosphere and the thermosphere. Despite its relative thinness and remoteness, the MLT plays a pivotal role in shaping our planet's climate and atmospheric dynamics (Smith,





2012b). It serves as a conduit for energy and momentum transfer, influencing global circulation patterns, tides, and planetary (PW) and gravity wave (GW) propagation (Smith et al., 2010; Becker, 2012). Moreover, the MLT region is linked to phenomena such as noctilucent clouds (Kuilman et al., 2017), travelling ionospheric disturbances (Hocke and Tsuda, 2001; Günzkofer

et al., 2023), and various space weather effects (Schrijver et al., 2015). Understanding the MLT's behavior is essential for comprehending broader atmospheric processes and their impact on Earth's environment and technological systems.

The coupling or interactions between the lower atmosphere (troposphere and stratosphere) and the MLT region has been tackled by many studies (Smith, 2012a; Becker, 2012). Ozone-hole-related effects represent one of the coupling mechanisms. Ozone losses in the springtime Antarctic lower stratosphere result in local cooling trends through reduced absorption of solar

radiation. Via thermal wind balance, the local cooling is associated with a strengthening of the polar vortex and a delay in its break-up. It has been established that zonal wind (ZW) trends were driven by ozone depletion in the Antarctic stratosphere due to emissions of ozone-depleting substances (Sun et al., 2014; Banerjee et al., 2020; Zambri et al., 2021). Model studies by Smith et al. (2010); Lossow et al. (2012); Lubis et al. (2016) have shown that the stratospheric changes affected the propagation of GWs from the troposphere to the mesosphere: the eastward GWs that drive the summer residual circulation cannot propagate

under these conditions of persisting ZW. Additionally, Ramesh et al. (2020), using the most recent WACCM6 simulations and a multiple linear regression model, showed that a negative trend in the stratosphere ZW after the year 2000 can be attributed to ozone recovery as suggested by Banerjee et al. (2020); Zambri et al. (2021).

Reanalyses have usually been used to analyse coupling processes only between the stratosphere and the troposphere due to their limited vertical extent and lack of assimilated observations in those regions. They are generally in good agreement in their

representation of the strengthening of the lower stratospheric polar vortex during the austral spring-summer season, associated with reduced radiative heating due to ozone loss (Orr et al., 2021). Wright and Hindley (2018) suggested that even though reanalyses assimilate the full suite of observations, they may be over-tuned to match their comparators. Fujiwara et al. (2022) report that variability among reanalyses grows with altitude. They recommended the use of multiple reanalyses in the upper stratosphere and lower mesosphere and warned that large discontinuities may occur due to differences in the data assimilation

process, which may preclude trend studies based on a single reanalysis system.

Historically, the ground-based observations of MLT winds have been carried out by medium-frequency (MF) radars and/or meteor radars (MRs). Long-term observations of winds at about 90 km height using non-height-resolving MR have been performed at Molodezhnaya (68°S, 46°E) since the 1960s until 1991. Trend analyses of the meridional wind until 1986 showed a decrease in magnitude during some months (Portnyagin et al., 1993; Bremer et al., 1997). They also reported a decrease in

the ZW, which was later confirmed by Merzlyakov et al. (2009) who added MF observations from Mawson (68°S, 63°E) since 1984. They showed, however, that there was a break towards positive ZW trends in the summer around 1990. Changes of zonal and meridional wind trends around 1990 have been observed at other latitudes also (Portnyagin et al., 2006). One may conclude that trends before 1990 cannot be compared to those detected from later observations. The Mawson radar was later moved to Davis (69°S, 76°E), where operations began in 1994. Since the 1990s, the continuous measurements in the southern

hemisphere have allowed us to compile climatologies of the MLT winds derived using the MF radars at Davis Station during 1994–2005 and those at Syowa Station during 1999–2003 (Dowdy et al., 2007). Additionally, long-term observations of the





mesopause region OH rotational temperature allowed trend and overall variability assessment at Davis (French et al., 2020b, a). Measurements at Syowa from 1999 to 2010 reveal a positive ZW trend in winter confined below 85 km in December and extending to higher altitudes into January, February, and March. Thus, these trends, to the extent that they are significant,
indicate longer-term variations of mean winds which can be resolved by existing data (Iimura et al., 2011). Recently, Noble et al. (2024) have studied the interannual variability of winds in the MLT over Rothera using MR in comparison with WACCM-X. Additionally, they have provided a comprehensive discussion and review on trends with time.

Stober et al. (2021b) compared observations from six meteor radars at various latitudes with results from three whole-atmosphere models, providing valuable insights into how mesospheric winds and tides differ between the hemispheres over
several years. As a complement to this study, we analyze long-term trends in MLT winds of meteor radar observations (see Section 2.1) and two whole-atmosphere models (see Section 2.2) in the southern hemisphere to examine the translated effect of ozone recovery observed in the stratosphere into the MLT region. We also check the compatibility of trends with two state-of-the-art reanalysis datasets (see Section 2.3.1). Section 3 describes MR data processing and our approach to trend analysis. The result section 4 splits into trend comparison among stations (Section 4.1) and trends related to GWs (Section 4.2); and trend
comparison between stations and models (Section 4.3). Finally, we conclude with a summary of the results, a discussion on the mechanism linking ozone recovery due to the Montreal protocol, and the long-term trends in MLT winds observed by MRs.

## 2 Data

### 2.1 Meteor radar observations

In this study we present long-term observations of three meteor radars located in the southern hemisphere: Davis (68.6°S,
78.0°E; Holdsworth et al., 2008), Rio Grande (53.7°S, 67.7°W; Fritts et al., 2010; de Wit et al., 2017), and Rothera (67–68°S, 68°W; Sandford et al., 2010; Dempsey et al., 2021). As shown in Fig. 1, Davis and Rothera are located at the same latitude, similarly, Rio Grande and Rothera are located at the same longitude. While all stations were combined for the common period 2008–2019 (see Section 4.1), Davis and Rothera, and Rio Grande are compared with models for the period 2005–2017, and 2008–2017, respectively (see Section 4.3).

GWs obtained from our methodology have periods of 5-10 min up to 1 h , horizontal wavelengths from 20-30 km up to 300-500 km, and vertical wavelengths of more than 5 km. Comparing radar-derived momentum fluxes to satellite estimates is not straightforward. As outlined in detail in Hocking (2005) and Stober et al. (2021a), meteor radars provide information on all Reynolds stresses and their directionality, while satellite-based estimates based on a single track of observations, like for SABER, provide only information on the magnitude without vector information. The observational filters of radar and SABER
for GWs are quite different because the radar estimates include GWs with horizontal wavelengths up to 500 km (Hocking, 2005; Wright and Hindley, 2018; Fritts et al., 2010; de Wit et al., 2014, 2017), while SABER, like most limb sounders, is sensitive to GWs of "true" horizontal wavelengths longer than about 200 km, or intrinsic wave periods longer than about 2 hours (see Fig. 8 in Alexander et al., 2010). The sensitivity is better for longer horizontal wavelengths. Further, the sensitivity to "true" horizontal wavelengths is better if a GW is viewed parallel to its wave fronts, and the apparent horizontal wavelength parallel



**Figure 1.** Map in Orthographic projection with meteor radar locations shown as bold crosses: Davis ($68.6°$S, $78.0°$E), Rio Grande (($53.7°$S, $67.7°$W; Fritts et al., 2010) and Rothera ($67–68°$S, $68°$W; Murphy et al., 2006; Sandford et al., 2010), and color-coded topography. Concentric circles represent latitudes of $50, 60, 70, 80°$S. In the vicinity of the meteor radars using a 300 km radius, SD-WACCM-X grid points are plotted in red.

to the line-of-sight becomes very long. For a detailed discussion of the SABER measurement geometry, see Trinh et al. (2015), and for an illustration of the different wavelengths, see, for example, Fig.2 in Ern et al. (2018).



## 2.2 Whole-atmosphere models

### 2.2.1 GAIA

Ground-to-topside Atmosphere and Ionosphere for Aeronomy (GAIA) is a global fully-coupled atmosphere circulation model including the Earth's troposphere, stratosphere, mesosphere, thermosphere, and ionosphere in the altitude range from the ground to ∼600 km for the neutrals and up to 3000 km for the plasma (Jin et al., 2012). The GAIA simulations have a horizontal resolution of $2.8° \times 2.8°$ (latitude×longitude) and a vertical resolution of 0.2 scale heights. The model uses parameterizations to account for GWs, with formulations by McFarlane (1987) for orographic GWs and by Lindzen (1981) for non-orographic GWs. In the troposphere, stratosphere, and mesosphere, a full radiation scheme developed by Nakajima et al. (2000) is used. The simulated atmospheric parameters (e.g., wind, temperature) are given in hourly values. A nudging technique is used to constrain the model output (pressure, temperature, wind, etc.) below 30 km altitude to the reanalysis data JRA-25/55 by the Japan Meteorological Agency with a $1.25° \times 1.25°$ spatial resolution and a 6-hour temporal resolution (Onogi et al., 2007; Kobayashi et al., 2015). Due to the update of JRA-25 to JRA-55 in 2014, the simulation uses JRA-55 for 2014-2016 and JRA-25 before that. The F10.7 index as a proxy for the EUV input was set to observed values, while a fixed cross-polar cap electric potential of 30 kV and almost no particle precipitation conditions, corresponding to low geomagnetic activity, were held throughout the simulation period.

### 2.2.2 SD-WACCM-X

Whole Atmosphere Community Climate Model Extension (WACCM-X) is an altitude-extended configuration of the Community Earth System Model (CESM; Hurrell et al. (2013)). WACCM-X models the whole atmosphere from the lower boundary (representing ocean, land, or ice) to the upper boundary in the thermosphere. Representation of the atmospheric physics in WACCM-X up to the lower thermosphere (∼ 130 km altitude) is similar to that of the conventional WACCM configuration (Marsh et al., 2013), while representation of the ionospheric physics is similar to the Thermosphere-Ionosphere Electrodynamics General Circulation Model (Richmond et al. (1992); Maute (2017)). Further details of the WACCM-X implementation are described by Liu et al. (2018). The Specified Dynamics (SD-WACCM-X) simulation run (Gasperini et al., 2020) used in this study constrains atmospheric dynamics up to ∼ 50 km altitude with reanalysis based on the Modern-Era Retrospective Analysis for Research and Applications (MERRA; Rienecker et al. (2011)) with the nudging procedure described in Smith et al. (2017). The simulated atmospheric dynamics are given with 3-hour time resolution on pressure levels with 1/4 scale height vertical resolution above the upper stratosphere, and uniform horizontal resolutions in latitude and longitude of 2.5° and 1.9°, respectively. The effects of non-orographic gravity waves (GWs) are parameterized using the source-oriented parameterization approach (Garcia et al., 2017), while orographic GWs are parameterized according to McFarlane (1987). The effects of geomagnetic activity at high latitudes are parameterized according to the planetary Kp index using the plasma convection model by Heelis et al. (1982).



## 2.3 Complementary datasets

### 2.3.1 MERRA2 and ERA5

We use MERRA2 (Modern Era Reanalysis for Research and Applications version 2, developed by NASA; Molod et al., 2015; Gelaro et al., 2017) and ERA5 (fifth generation of ECMWF reanalysis; Hersbach et al., 2020) reanalysis datasets to complement the radar measurements below 80 km. MERRA2 and ERA5 were analyzed on model levels and on a 3-hourly and hourly basis, respectively. MERRA2 has a native resolution of 0.5° in latitude and 0.625° in longitude, and 72 hybrid sigma/pressure levels, extending to 0.01 hPa. ERA5 is available on a 31km grid (0.25° x 0.25°) and resolves the atmosphere using 137 levels from the

surface up to a height of 80km. Similarly to whole-atmosphere models, we include only grid points in the vicinity of the meteor radars using a 300 km radius (see Fig. 1). While discontinuities in the upper atmosphere could exist in 1979, 1985, 1998, and 2004 (McLandress et al., 2014; Kuchar et al., 2015; Simmons et al., 2020), coinciding with major changes in instrumentation or analysis procedure, no discontinuities have been reported for our covered period. A comparison of trace gas volume mixing ratio between MERRA2 and ground-based and space-borne remote sensing can be found in Shi et al. (2023, 2024). MERRA2

leverages a simplified ozone chemistry approach that is affected by odd oxygen chemistry, resulting in large discrepancies in the ozone volume mixing ratio at polar latitudes during polar night conditions at the lower mesosphere compared to radiometric trace gas data, measured with ground-based or space-borne instruments such as MLS.

### 2.3.2 SABER data

We analyse Sounding of the Atmosphere using Broadband Emission Radiometry (SABER) GW data for the period 2002–2022.

Since SABER data are very sparse, we averaged over 5-day intervals and over 30° longitude and 12° latitude centered at about the location of the respective station: Davis (65°E–95°E, 75°S–63°S), Rio Grande (85°W–55°W, 60°S–48°S) and Rothera (85°W–55°W, 74°S–62°S). The Thermosphere Ionosphere Mesosphere Energetics and Dynamics (TIMED) satellite, carrying the SABER instrument, performs yaw maneuvers every about 60 days, which means that the SABER latitude coverage changes accordingly. For northward-viewing periods the SABER latitude coverage is about 50°S-82°N, and 82°S-50°N for southward-

viewing periods. As a consequence of the changes in latitude coverage, there are gaps in the GW time series for Rothera and Davis. For details about the extraction of GW data from SABER observations we refer to Ern et al. (2018).

We focus on SABER GW potential energy per volume ($E_{pot,V}$), given in $J\,m^{-3}$, since GW momentum fluxes are generally quite noisy and only a fraction of the data can be used for calculations. $E_{pot,V}$ also contains density and is therefore closer to GW momentum flux than GW potential energy per unit mass. If GW horizontal and vertical wavelengths do not change,

$E_{pot,V}$ should be a conserved quantity, like GW momentum flux (Fritts and Alexander, 2003). Changes of the background wind with altitude, however, will cause Doppler shifts, which in turn would lead to changes in $E_{pot,V}$, while GW momentum flux may still be conserved. $E_{pot,V}$ has the advantage that seasonal variations of atmospheric background density at a fixed altitude are accounted for, and seasonal variations of GW activity and of the background wind can be better separated.





## 3 Methodology

### 3.1 Adaptive spectral filtering

Atmospheric mean winds and tides are analyzed using the adaptive spectral filter (ASF2D), which is described in more detail in Baumgarten and Stober (2019) and Stober et al. (2020) and was already applied in several studies (Stober et al., 2017; Pokhotelov et al., 2018; Stober et al., 2021b) to decompose MR winds to daily mean winds, diurnal and semidiurnal tides for the zonal and meridional components. The main advantage of the technique lies in the robustness of the fitting method that permits the decomposition of time series with unequal sampling (irregular time grid) and data gaps. Furthermore, the method

provides information about the statistical uncertainties of the obtained Fourier coefficients. The ASF2D method was already used for a meteor radar wind trend study presented in Wilhelm et al. (2019). Based on the decomposed time series, long-term changes in mean winds, tides, and PWs were analyzed for almost two decades using meteor radars in the northern hemisphere.

### 3.2 Trend analysis

Linear trends for zonal and meridional winds have been estimated from monthly means calculated from the preprocessed original data using ASF2D by the Theil–Sen estimator. The Theil–Sen estimator, proposed by Theil (1950) and Sen (1968) to estimate the magnitude of the monotonic trend, has been widely used to assess linear trends due to its insensitivity to outliers. The breakdown point is 0.29, meaning that its accuracy is robust up to 29% of corrupted input data points (Wilcox, 2001). Additionally, it has high asymptotic efficiency compared to the least-squares estimator[1] when the error term is heteroscedastic

and normally distributed, or when homoscedastic and heavy-tailed (Wilcox, 1998).

The Hamed and Rao Modified Mann-Kendall (MK) test (Hamed and Ramachandra Rao, 1998) has been used to assess the significance level (p-value) of our trend estimates to address serial autocorrelation issues. Other modifications of the MK test, e.g., taking into account the 11-year solar cycle as an external driver (Libiseller and Grimvall, 2002), have been tested and did not produce significantly different results.

## 170 4 Results

### 4.1 Trend comparison among stations

Figure 2 shows the trend intercomparison of zonal (upper panels) and meridional (lower panels) monthly mean winds at the locations of Davis, Rio Grande, and Rothera for the common period 2008–2021. Meteor radar measurements are complemented by MERRA2 below 80 km. The equivalent figures using ERA5 (see Fig. A1 in Appendix) show similar findings but with lower

magnitude. The most striking is the common negative trend in stratosphere/mesosphere ZW starting in September, causing a delay in the vortex transition from westerlies to easterlies in the stratosphere and lower mesosphere. This negative trend has been attributed to ozone recovery after 2000 in the southern hemisphere (Sun et al., 2014; Banerjee et al., 2020; Zambri et al.,

---

[1]Estimators with low efficiency require more independent observations to attain the same sample variance of efficient unbiased estimators.





2021). Similarly to Venkateswara Rao et al. (2015), we present evidence that the stratospheric and mesospheric wind strengths are highly anti-correlated and strongly affected by the perturbations induced by the Antarctic ozone hole. The zonal winds at 30 km in November (see Fig. A2) show positive correlations with the ozone hole area ($r \geq 0.55$) at all locations. Fig. A3 shows the relation between the mesospheric zonal winds at 82 km and ozone hole area in November at all locations and reveals even higher correlations ($r \geq 0.63$). We attribute this opposite relation to GWs discussed in Section 4.2.

The negative trend is preceded at similar heights by a positive trend revealed at the locations of Davis and Rio Grande during the austral winter months, and also in agreement with Banerjee et al. (2020). This trend is less pronounced at Rothera and is rather presented as a negative trend starting in March. A significant positive trend in ZWs is also reported near the stratopause over the King Sejong Station (Song and Song, 2024).

Trends between 70–80 km are consistent with the trends revealed above by MRs. This serves as a weak check of the relevance of MERRA2 (and ERA5) in the mesosphere and as a compatibility check between MRs and reanalyses. The common negative trend in ZW starting in September mentioned above switches to a positive trend, weakening easterlies between 70–90 km. This appears in agreement with the trend estimate over Rothera by Noble et al. (2024) using a different trend analysis technique.

The lower panels of Fig. 2 show trend intercomparison of meridional winds. The patterns are not as clear as in the case of ZW, in part because the amplitudes of meridional wind trends are more than twice less than ZW. Therefore, it is quite difficult to find a common trend structure. One common pattern between Rio Grande and Rothera is the negative trend that started in August around 60 km, further strengthening the southward circulation in this region. This negative trend repeats above 80 km in the case of Rio Grande and above 90 km in the case of Rothera. The negative trend also repeats below 40 km. In the case of Davis, the southward circulation is strengthened and weakened below and above 45 km, respectively. Another common pattern is the weakening of northward circulation in austral summer, estimated above and below 90 km at Rothera and Davis, respectively. This trend pattern changes at the location of Rio Grande, where we find a positive trend in Nov/Dec, i.e., strengthening of northward circulation. Furthermore, we note that the meridional trends do not reveal the same level of compatibility between MRs and reanalyses compared to the zonal wind due to a weaker observational constraint than the thermal-wind constraint on zonal flow (Fleming and Chandra, 1989; Martineau et al., 2016).

Similarly to our analysis at Davis, positive trends in meridional winds below 90 km in the austral summer were reported by Vincent et al. (2019) using MF radar wind measurements made at Davis. The enhanced meridional circulation is linked with an increased upwelling and may result in colder temperatures close to the summer mesopause due to adiabatic effects and vice versa (Smith et al., 2010; Ramesh et al., 2020).

## 4.2 Trends related to GWs

The common positive trend in ZW around 80 km starting in September is robust across stations and also between reanalyses and observations. Thus, we zoom in on the trend estimate of zonal monthly mean winds captured by MRs (see Fig. A3). The ZW trend feature in spring/early summer at 80 km corresponds with the positive trend estimate in zonal momentum flux as a proxy for GWs revealed at Davis and Rothera (see Fig. 3**A** and **C**). The trend estimate at Rio Grande does not indicate any significant changes in zonal momentum flux (see Fig. 3**B**). The weakening and strengthening of westward and eastward




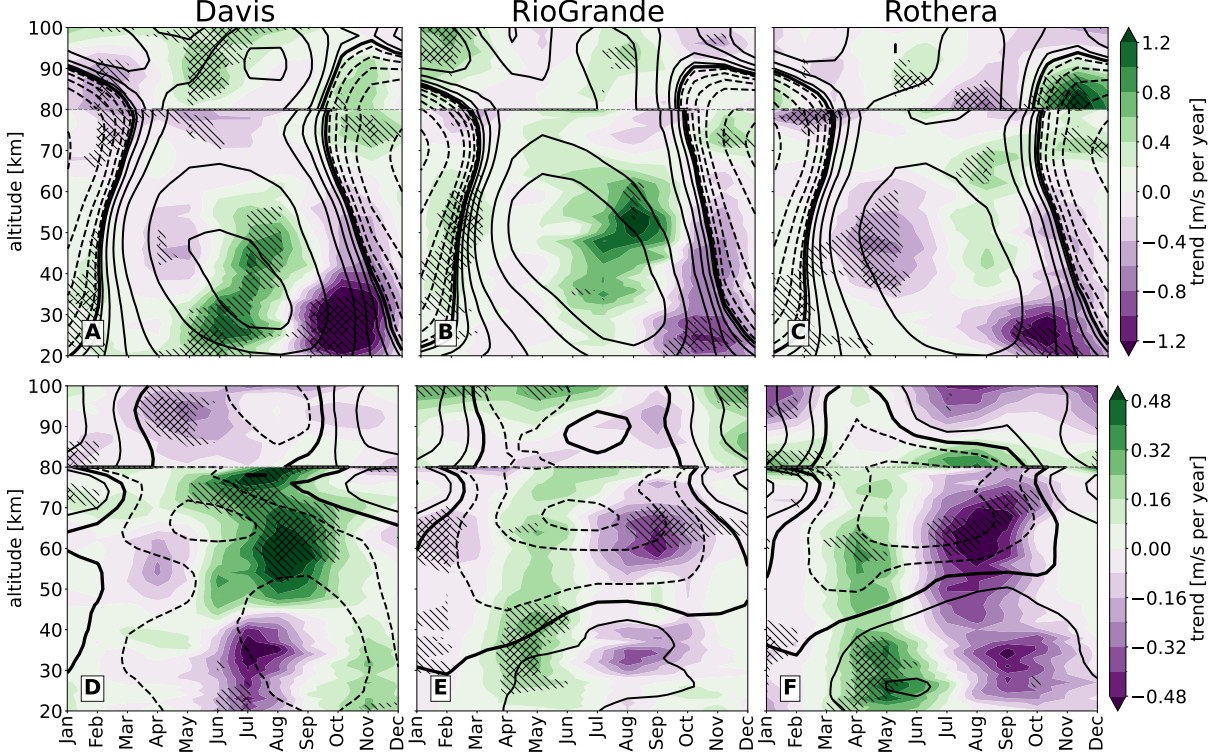

**Figure 2.** Trend comparison (shading) of zonal (upper panels) and meridional (lower panels) monthly mean winds at the locations of Davis, Rio Grande, and Rothera for the common period 2008–2021. MERRA2 and meteor radar measurements cover the altitude ranges 20–80 km and 80–100 km, respectively. Solid (positive values) and dashed (negative values) contours represent wind climatology with the following levels: $0, \pm 2, \pm 6, \pm 12, \pm 20, \pm 40, \pm 60$ m/s. Hatching \\\\ and //// shows where the p-values of the MK test are $< 0.05$ and $< 0.01$, respectively.

zonal momentum flux at Davis and Rothera, respectively, starts in October below 90 km when westerlies in the stratosphere weaken (see Fig. 2). While at Davis it may thus filter the westward zonal momentum being carried even from the troposphere, at Rothera the eastward zonal momentum flux is stronger below 90 km (see Fig. 3C). Similar to Davis, the strengthening of

215 the net eastward momentum flux at Rothera should also be an effect of a reduction of the westward momentum fluxes in the overall spectrum of gravity waves at that altitude. Above 90 km we observe strengthening of the westward zonal momentum flux. These features indicate the coupling via GWs between ozone recovery effects in the stratosphere and the MLT, which vary on a longitudinal and latitudinal basis.

Similarly, the ZW trend in January and February reveals a delayed transition from summer (easterly) to winter (westerly)

220 circulation. This is associated with weaker westward and stronger eastward momentum flux below 90 km at the locations of Davis and Rothera, respectively. At Rothera, strengthening westward momentum flux above 90 km is persistent throughout the year.





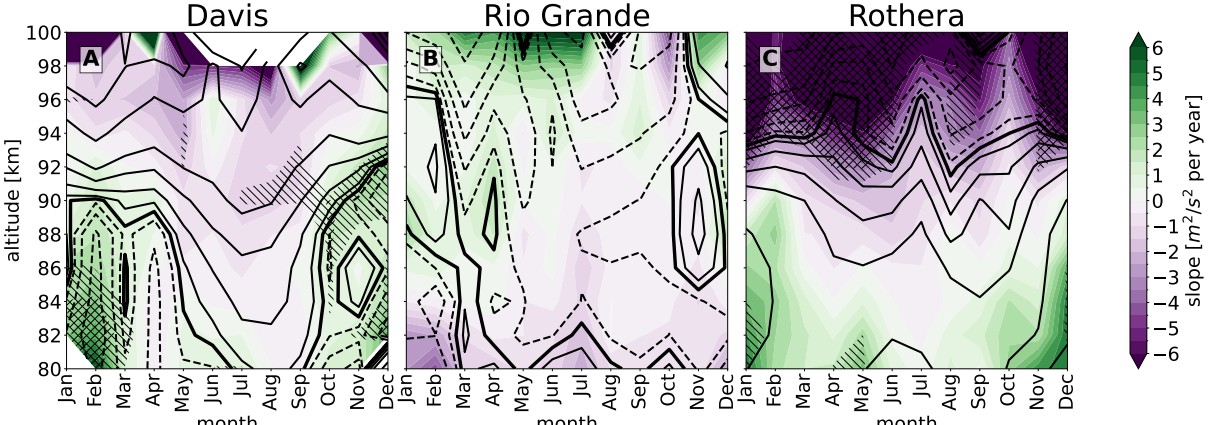

**Figure 3.** Trend (shading) of zonal momentum flux (MF) at the location of Davis (**A**), Rio Grande (**B**) and Rothera (**C**) for the period 2005–2021, 2008–2021 and 2005–2021, respectively. Solid (positive values: $\{2, 6, 12, 20, 40, 60\}\,\mathrm{m^2\,s^{-2}}$) and dashed (same as positive but with negative signs) contours represent MF climatology. Hatching \\\\\ and //// shows where the p-values of the MK test are $< 0.05$ and $< 0.01$, respectively.

To examine whether the trend in the time series of the zonal momentum at the location of Davis in December (see Fig. 4) is spurious or real, we apply several trend techniques (see Fig. 5A). Switching from Theil-Sen estimates to ordinary (OLS) and generalized (GLS) least squares, we receive lower standard errors when compared to GLS with autoregression covariance structure (GLSAR) and using bootstrap resampling on OLS (see **OLS (boot)** in Fig. 5) and Theil-Sen estimates (see **Theil-Sen (boot)** in Fig. 5). **Theil-Sen (boot)** reveals as the one with the largest confidence intervals due to its non-normal (double-peaked) slope distribution (see Fig. 5B). The use of violin plots allows us to assess that the trend in zonal momentum flux is robust even when Theil-Sen and bootstrap techniques are combined.

Alternatively, we may pose a question of whether the time series of zonal momentum flux at the location of Davis and 82 km in December from Fig. 4 is long enough for the period 2005–2021 to emerge a significant trend. We use the approach of Sledd and L'Ecuyer (2021) to assess time to emerge (TTE) in years in Fig. 6. We create a synthetic time series by generating random noise with the same variance and autocorrelation of the detrended time series, and to this noise, we add the linear trend determined from the original time series. Analytical approaches by Weatherhead et al. (1998) and Leroy et al. (2008) have a long history of climate change detection at the surface (Phojanamongkolkij et al., 2014). We document that the given time series is long enough, i.e., fewer years ( 8 years) are needed, to assess a robust trend estimate.

### 4.2.1 Comparison with satellite estimates

We discuss whether the satellite-based GW estimates can assist in revealing trends, similar to the way in which the reanalyses complement the MRs as shown above. The radar estimates include GWs with horizontal wavelengths up to 500 km, compared to SABER being more sensitive to GWs with horizontal wavelengths above this scale. Ern et al. (2018) documents an increase



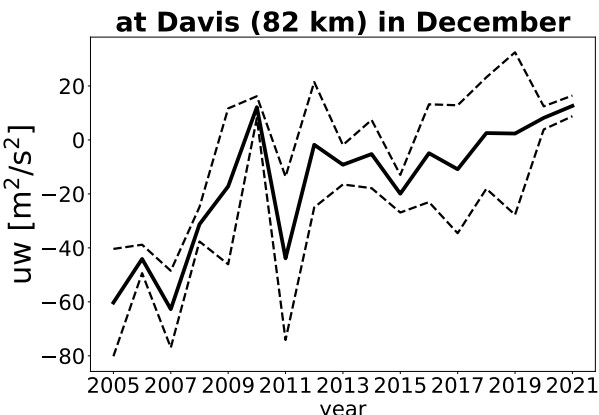

**Figure 4.** Zonal momentum flux (MF) time series $[\mathrm{m}^2\,\mathrm{s}^{-2}]$ at the location of Davis at 82 km in December from Fig. 3A. Dashed lines represent uncertainty estimates.

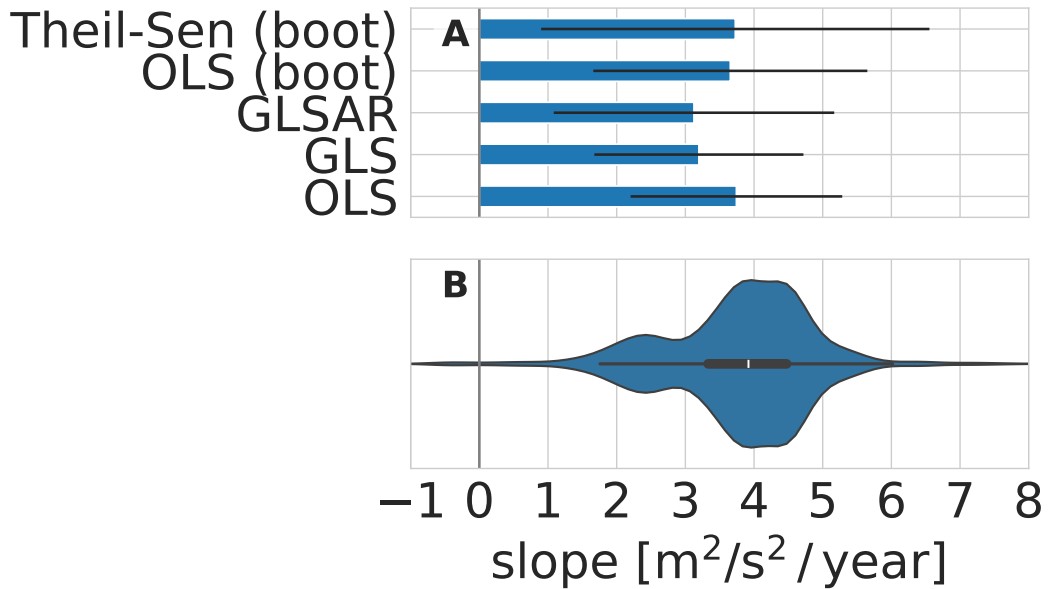

**Figure 5.** Method comparison of zonal momentum flux (MF) trend estimate at the location of Davis at 82 km in December (**A**) from Fig. 4. The comparison includes the following methods: ordinary least squares (OLS), generalized least squares (GLS), GLS with autoregression covariance structure (GLSAR), bootstrap resampling of OLS estimates (OLS (boot)), and bootstrap resampling of Theil-Sen estimates (Theil-Sen (boot)). The bootstrapped Theil-Sen estimates document a double-peaked slope of zonal momentum flux (**B**). The white line in the violin plot depicts the position of the median, and the horizontal black bar represents the quartile interval.

in horizontal wavelengths with altitude. SABER is sensitive to GWs of horizontal wavelengths longer than about 200 km, however, with better sensitivity at longer horizontal wavelengths (Alexander et al., 2010; Trinh et al., 2015).





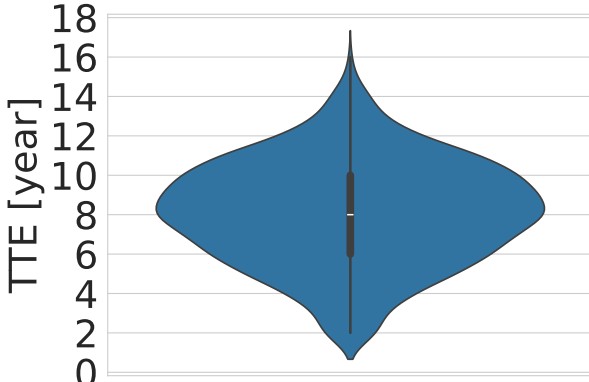

**Figure 6.** Is our time series from Fig. 4 long enough? Time to emergence (TTE) is the expected number of years needed to detect a trend with a 95% confidence level. The white line in the violin plot depicts the position of the median, and the vertical black bar represents the quartile interval.

Fig. 7 shows a trend comparison (shading) of monthly averaged GW potential energy ($E_{pot,V}$) at all three locations. We omit time series with limited temporal coverage to avoid spurious trends (white spaces). Davis and Rothera as high-latitude stations,
245 do not fulfill our condition for temporal coverage in February, June, and October. We document similarly slightly positive trends during austral fall/early winter, indicative of stronger $E_{pot,V}$ in the strengthening eastward-directed polar vortex winds at the same altitude, and negative trends in about Nov/Dec corresponding to the weakening winds in these months. The negative trend starting in November above 80 km at all stations corresponds with the weakening of the zonal momentum flux indicated in Figs. 3, 4, 5, and 6. The negative trends in Nov/Dec are stronger at Davis in terms of magnitude.
250 Regarding the comparison with radar momentum fluxes, it should be mentioned that the radar momentum fluxes in Fig. 3 are net momentum fluxes, which means that there can be cancellation effects making a direct comparison with SABER more difficult. In particular, the positive trend in Rothera in November/December below 90 km should be caused by a weakening of westward momentum fluxes because of the wave filtering effect by the more westward wind in the stratosphere. This could result in a weakening of absolute momentum fluxes at 80–90 km, which would correspond to the negative trend of SABER
255 Epot in these months in Fig. 7**C**.

### 4.3 Trend comparison between observations, reanalysis, and models

In the following subsections, we compare trend estimates in zonal and meriodional winds between MRs, MERRA2 (for ERA5 see Figs. A4,A5,A6), and models (GAIA and SD-WACCM-X) at the locations of Rio Grande, Davis and Rothera for various periods.





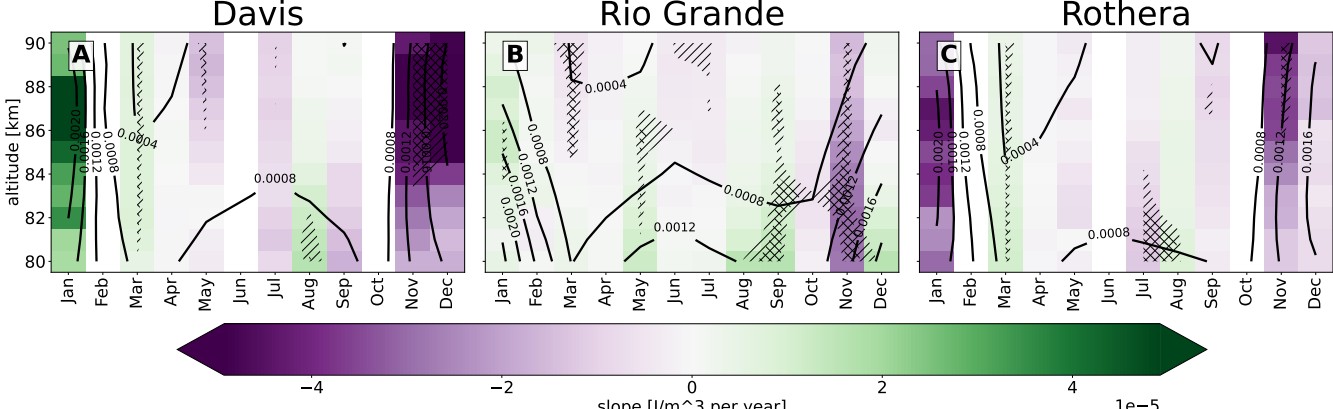

**Figure 7.** Trend comparison (shading) of monthly averaged GW potential energy ($E_{pot,V}$) at the locations of Davis (**A**), Rio Grande (**B**), and Rothera (**C**) for the common period 2002–2022. Hatching \\\\ and //// shows where the p-values of the MK test are $< 0.05$ and $< 0.01$, respectively. Regions of limited temporal (less than 10 years) coverage are left blank. Solid contours represent $E_{pot,V}$ climatology.

### 4.3.1 Rio Grande for the period 2008–2017

Figure 8 shows the trend intercomparison of zonal (upper panels) and meridional (lower panels) monthly mean winds measured at the locations of Rio Grande and simulated with SD-WACCM-X and GAIA for the common period 2008–2017. While the common negative trend in ZW starting in September in the stratosphere and lower mesosphere is well reproduced by the GCMs, its vertical extent is better captured by SD-WACCM-X, probably since the simulation constraint goes higher, i.e., up to $\sim 50\,\mathrm{km}$, compared to GAIA. The positive trend above $70\,\mathrm{km}$, also documented in Fig. 2, is better captured by GAIA between 80 and $90\,\mathrm{km}$. SD-WACCM-X reveals the negative summer trend in the region of ZW reversal below $80\,\mathrm{km}$. SD-WACCM-X and GAIA show that the summer ZW reverses lower and higher, respectively, compared with the radar measurement at the location of Rio Grande, in agreement with Stober et al. (2021b). We note that the trends in this region are not statistically significant for the quite short period 2008-2017, considering our method.

The positive trend between 40 and 60 km in February and March, basically accelerating the transition between summer and winter circulation, is well reproduced by both models. During the winter months, we find positive trends accelerating westerlies in both models and observations. This trend is specific to the location of Rio Grande and does not appear at other locations.

Figs. 8**D–F** show that a negative trend in the meridional wind in JJA is only reproduced by GAIA, similarly to Rothera in Fig. 10**D–F**. This trend appears at all layers between 20 and 100 km, though it is statistically significant only at particular layers. The negative and positive trend at the beginning and the end of summer, respectively, is not captured by the GCMs.

### 4.3.2 Davis for the period 2005–2017

Figure 9 shows the trend intercomparison of zonal (upper panels) and meridional (lower panels) monthly mean winds measured at the location of Davis and simulated with SD-WACCM-X and GAIA for the common period 2005–2017. In contrast to



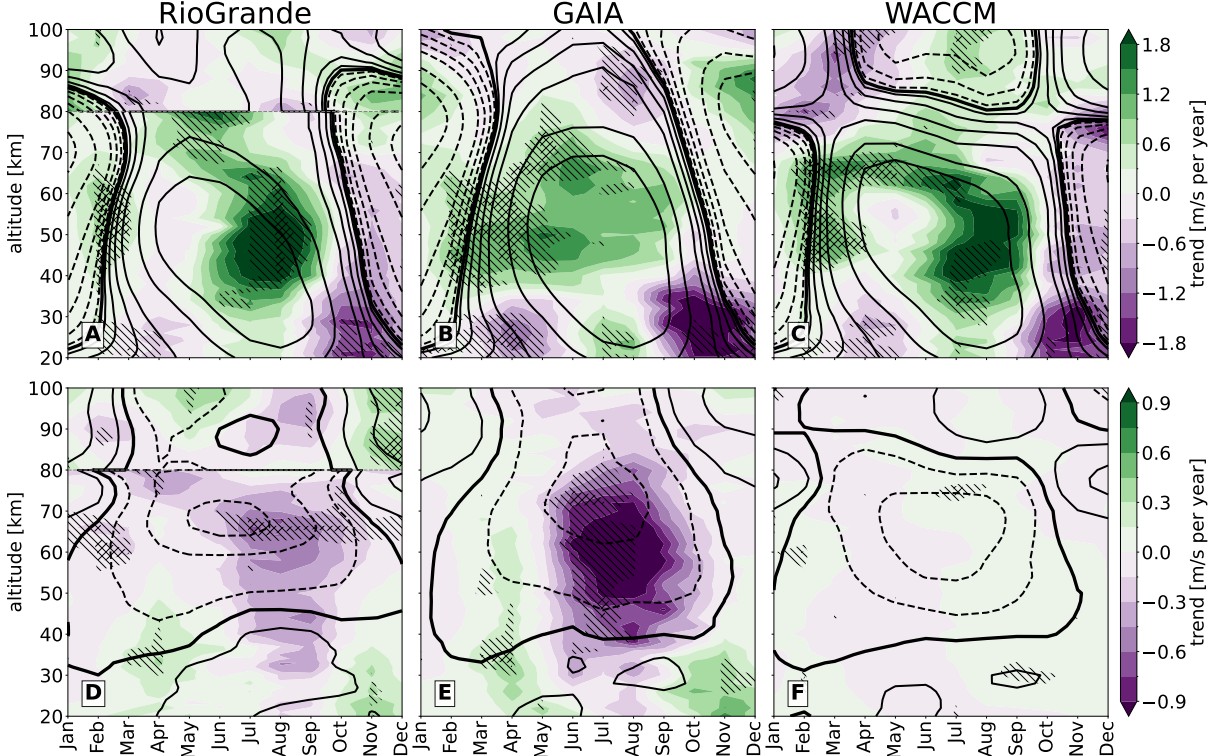

**Figure 8.** Trend comparison (shading) of zonal (upper panels) and meridional (lower panels) monthly mean winds at the locations of Rio Grande between reanalysis/observations (MERRA2 and meteor radar measurement for the altitude 20–80 km and 80–100 km, respectively; **A,D**), GAIA (**B,E**), and SD-WACCM-X (**C,F**). The comparison is for the common period 2008–2017. Contours represent wind climatology with the following levels: $\mathbf{0}, \pm 2, \pm 6, \pm 12, \pm 20, \pm 40, \pm 60 \, \mathrm{m/s}$. Hatching \\\\ and //// shows where the p-values of the MK test are $< 0.05$ and $< 0.01$, respectively.

Rio Grande (Fig. 8). Both models can reproduce the positive ZW trend in Nov/Dec around 80 km, even though the linked
negative trend in the stratosphere is vertically limited by 50 km in MERRA2 and 30 km in both models, respectively. While SD-WACCM-X reveals issues reproducing the basic climatology in winter (Stober et al., 2021b), the long-term wind changes are, in general, better reproduced.

Negative trends and positive trends in the meridional wind in Apr/May/Jun and Aug/Sep/Oct, respectively, are well reproduced only by GAIA (cf. Figs. 9**D** and 9**E**). While trend slopes in SD-WACCM-X indicate a stronger change compared to the
other two stations, it does not match meteor radar observations.



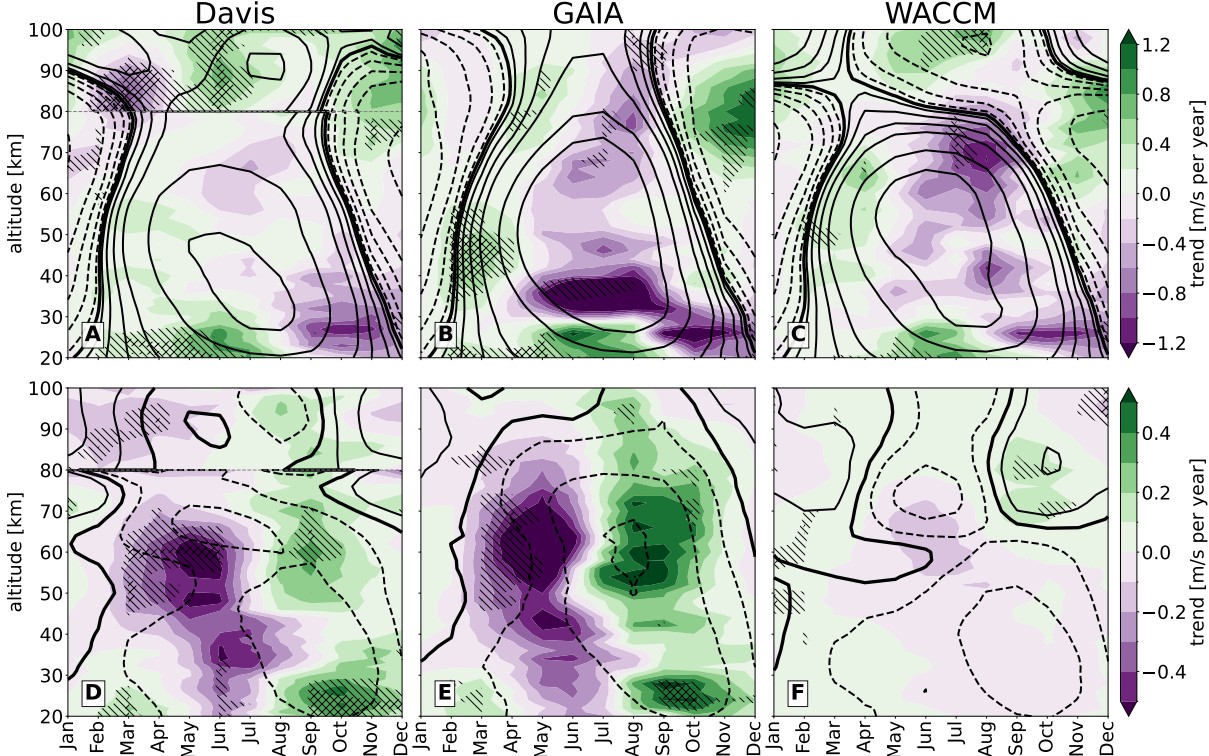

**Figure 9.** Trend comparison (shading) of zonal (upper panels) and meridional (lower panels) monthly mean winds at the locations of Davis between observations (MERRA2 and meteor radar measurement for the altitudes 20–80 km and 80–100 km, respectively; **A**,**D**) and GAIA (**B**,**E**) and SD-WACCM-X (**C**,**F**). The comparison is for the common period 2005–2017. Contours represent wind climatology with the following levels: $0, \pm 2, \pm 6, \pm 12, \pm 20, \pm 40, \pm 60$ m/s. Hatching \\\\ and //// shows where the p-values of the MK test are $< 0.05$ and $< 0.01$, respectively.

### 4.3.3 Rothera for the period 2005–2017

Figure 10 shows the trend intercomparison of zonal (upper panels) and meridional (lower panels) monthly mean winds measured at the locations of Rothera and simulated with SD-WACCM-X and GAIA for the common period 2005–2017. Similarly to Fig. 9, both models reflect the positive ZW trend in Nov/Dec around 80 km.

Negative trends and positive trends in the meridional wind in Apr/May/Jun and Aug/Sep/Oct, respectively, are well reproduced only by GAIA (cf. Figs. 10**D** and 10**E**). Not even stratospheric trend slopes in SD-WACCM-X agree with MERRA2 as they do in Fig. 8**F**. This is surprising since SD-WACCM-X is nudged towards MERRA2.



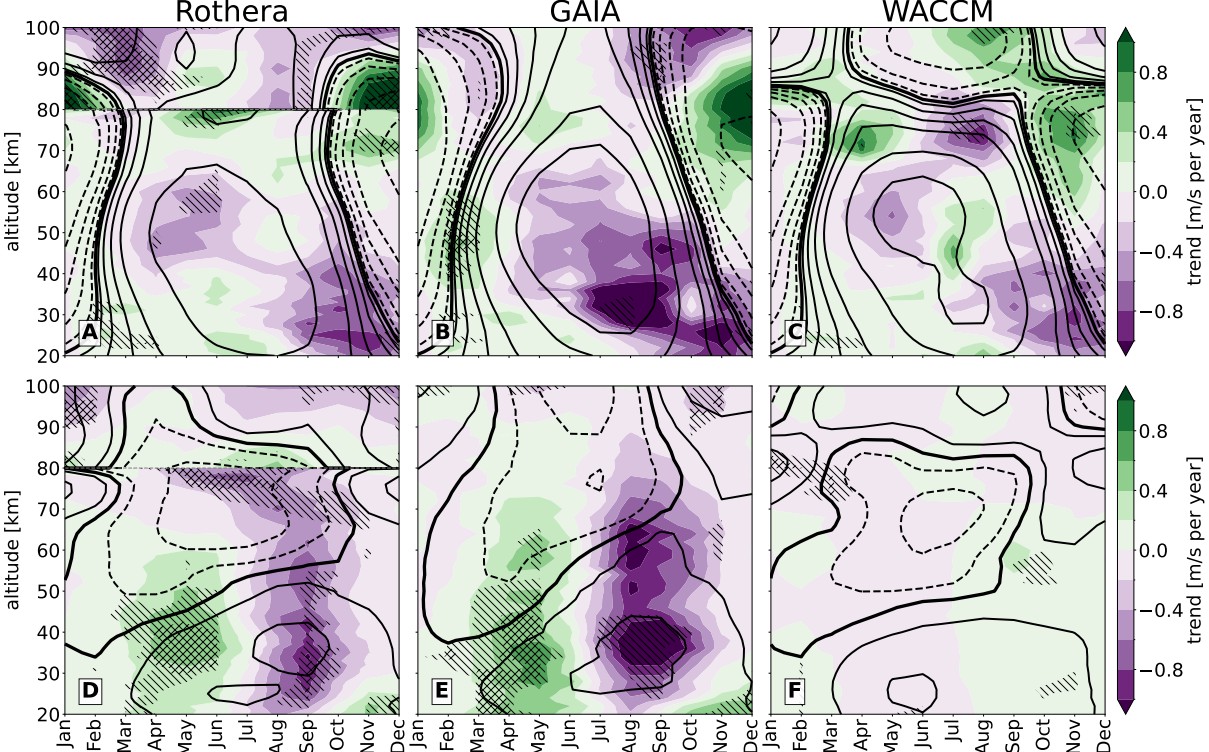

**Figure 10.** Trend comparison (shading) of zonal (upper panels) and meridional (lower panels) monthly mean winds at the locations of Rothera between observations (MERRA2 and meteor radar measurement for the altitudes 20–80 km and 80–100 km, respectively; **A,D**) and GAIA (**B,E**) and SD-WACCM-X (**C,F**). The comparison is for the common period 2005–2017. Contours represent wind climatology with the following levels: $0, \pm 2, \pm 6, \pm 12, \pm 20, \pm 40, \pm 60$ m/s. Hatching \\\\ and //// shows where the p-values of the MK test are $< 0.05$ and $< 0.01$, respectively.

## 5 Discussion and conclusions

Using long-term observations from meteor radar stations and simulations from two whole-atmosphere models we provide evidence for a causal chain linking stratospheric ozone recovery to changes in mesospheric dynamics in the Southern Hemisphere. The recovery of the Antarctic ozone layer has led to a weakening and earlier breakdown of the stratospheric polar vortex, particularly evident in the zonal winds near 30 km. These changes alter the filtering conditions for upward-propagating GWs. This filtering is manifested in both the positive trends in zonal winds around 80-–90 km and the trends in GW momentum flux and potential energy observed in meteor radar and SABER data, especially during austral spring and early summer. Thus, we relate mesospheric wind trends to stratospheric changes previously attributed to ozone recovery.

Our findings underscore the coupling between the stratosphere and the MLT region, facilitated by PWs, GWs of various scales, and by thermal tides. In particular, we explored the role of GWs using both MR-derived momentum flux and satellite-



derived potential energy ($E_{pot,V}$). These complementary diagnostics strengthen the evidence for vertical coupling via GWs and highlight the value of combining ground-based and satellite observations in future studies.

The analysis reveals a significant negative trend in ZWs starting in September. It suggests that the transition from westerlies to easterlies is delayed in the stratosphere. This trend is attributed to the recovery of ozone levels post-2000, as corroborated by previous studies (Sun et al., 2014; Banerjee et al., 2020; Zambri et al., 2021) and further supported by using the WACCM6 simulation and multiple linear regression models (Ramesh et al., 2020). While previous modelling studies of Smith et al. (2010) and Lubis et al. (2016) demonstrated the influence of stratospheric ozone changes on gravity wave (GW) propagation, which

directly affects mesospheric dynamics, the long-term connection of these observed mesospheric variations to changes in the springtime stratospheric ozone loss has not been explored. However, Venkateswara Rao et al. (2015) presented observational pieces of evidence of the influence of Antarctic stratospheric ozone variability on mesospheric wind using both mesospheric MF radar wind observations from Rothera as well as the MERRA reanalysis data. In our study, the observed long-term trends in mesospheric winds are consistent with the changes in stratospheric dynamics, highlighting previous suggestions of vertical

coupling via GWs.

Our results also indicate that the positive trend in zonal momentum flux around 80 km, starting in September, is robust across different MR stations, given the relatively short period of measurements. This trend corresponds to the weakening of westward zonal momentum flux, furthermore supporting a filtering effect of the stratospheric westerlies on the mesospheric dynamics. The ASF2D method employed in this study (Baumgarten and Stober, 2019; Stober et al., 2020) has proven effective

in isolating these long-term trends from the high variability inherent in atmospheric data.

The comparison between observations and model simulations reveals that, while both GAIA and SD-WACCM-X models have limitations in reproducing the basic climatology of the mesosphere (Stober et al., 2021b), GAIA shows a somewhat better agreement with the observed trends. This highlights the importance of continuous model validation and improvement, particularly in the representation of mesospheric processes and their coupling with the lower atmosphere.

Observational studies by Noble et al. (2024) and Dowdy et al. (2007), using meteor radar measurements, highlight the changes in mesospheric wind patterns and the impacts of stratospheric trends on the MLT dynamics. The common trends observed in both reanalysis and meteor radar data across several locations in the SH underscore the broader regional impact of ozone recovery on mesospheric dynamics, as illustrated through changes in zonal and meridional wind patterns. These findings affirm that the dynamics within the MLT region are influenced by stratospheric ozone recovery processes, as partly supported

by recent studies using whole atmosphere models like GAIA and WACCM-X. This emphasizes the role of vertical coupling via GWs, demonstrating the interconnected nature of atmospheric layers and the critical impact of ozone recovery on mesospheric dynamics.

We attribute our results as a consequence of the recovery of the Antarctic ozone layer, moderated by international policies such as the Montreal Protocol, and induced significant changes in atmospheric circulation patterns, extending their effects to

MLT through a complex coupling mechanism involving GWs. However, it is unclear how the ozone recovery impacts the MLT through other coupling agents such as tides, etc. This would need to be explored in future studies. A similar indicator that



the climate mitigation is taking effect could be found in the middle and upper stratosphere (Romanzini-Bezerra and Maycock, 2024).



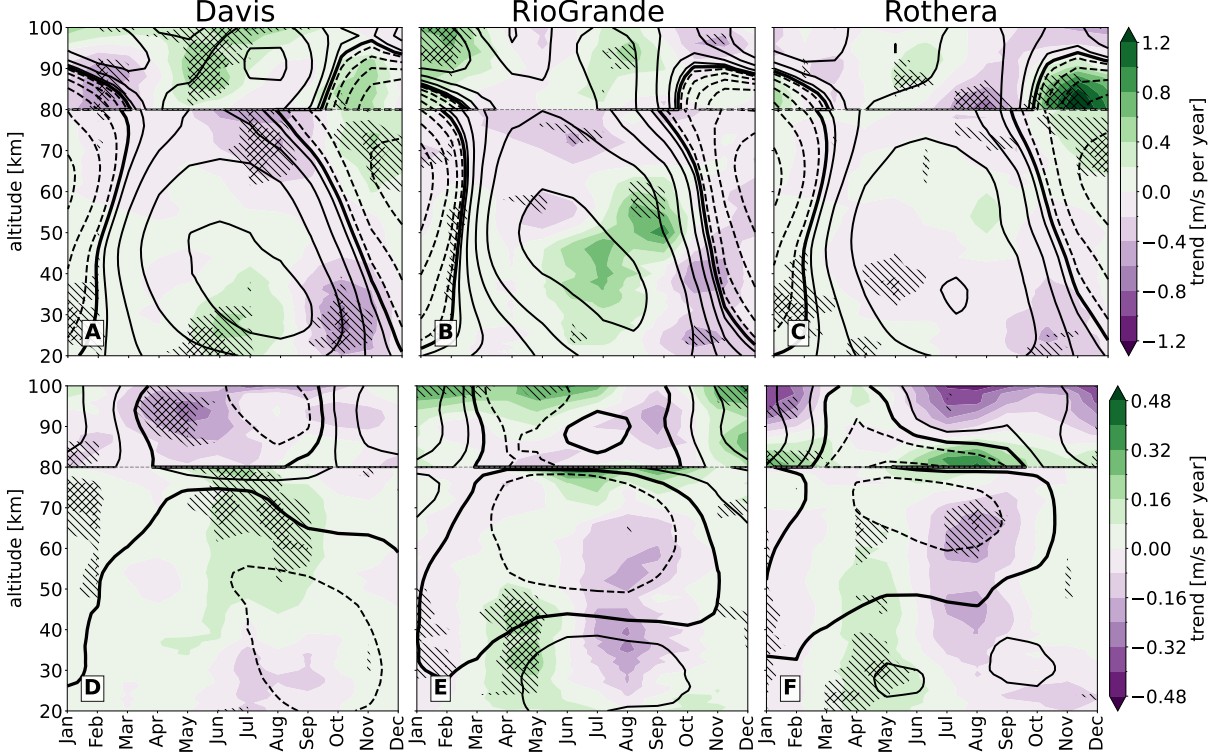

**Figure A1.** Trend comparison (shading) of zonal (upper panels) and meridional (lower panels) monthly mean winds at the locations of Davis, Rio Grande, and Rothera for the common period 2008–2021. As in Fig. 2 but in this case ERA5 and meteor radar measurements cover the altitude ranges 20–80 km and 80–100 km, respectively. Solid (positive values) and dashed (negative values) contours represent wind climatology with the following levels: $\mathbf{0}, \pm 2, \pm 6, \pm 12, \pm 20, \pm 40, \pm 60\,\mathrm{m/s}$. Hatching \\\\ and //// shows where the p-values of the MK test are $< 0.05$ and $< 0.01$, respectively.

# Appendix A: Additional results

Ozone-hole area and the total ozone mass deficit used in Figs. A3, A2 are taken from the NASA Ozone Watch (http://ozonewatch.gsfc.nasa.gov/meteorology/SH.html) and are derived from observations from a variety of satellites. The ozone hole area is defined to be that region of ozone values below 220 Dobson units (DU) located south of 40°S.



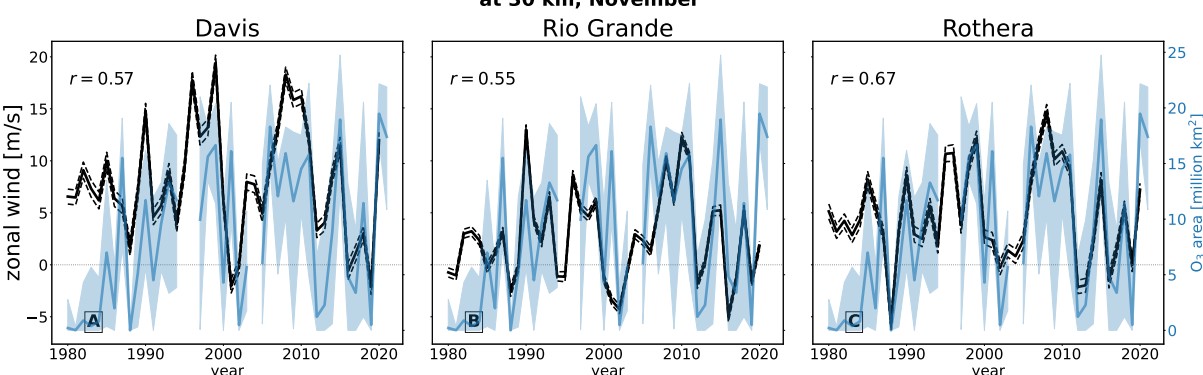

**Figure A2.** Zonal wind monthly time series [m/s] compiled for MERRA2 at the location of Davis (**A**), Rio Grande (**B**) and Rothera (**C**) at 30 km in November (black solid lines). Blue lines represent ozone hole area in the SH in million km$^2$. Dashed black lines and shading represent uncertainty estimates for particular time series. $r$ indicates the correlation coefficient between those time series.

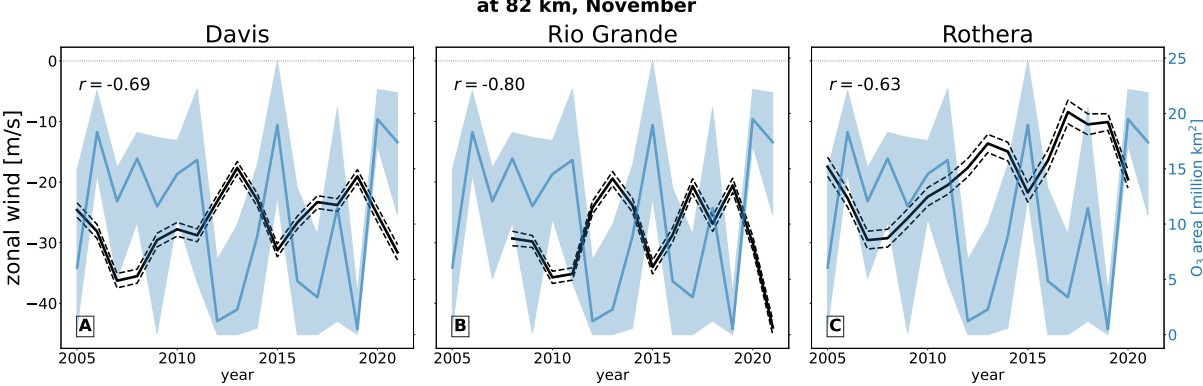

**Figure A3.** Zonal wind monthly time series [m/s] at the location of Davis (**A**), Rio Grande (**B**) and Rothera (**C**) at 82 km in November. Blue lines represent ozone hole area in the SH in million km$^2$. Dashed black lines and shading represent uncertainty estimates for particular time series. $r$ indicates the correlation coefficient between those time series.

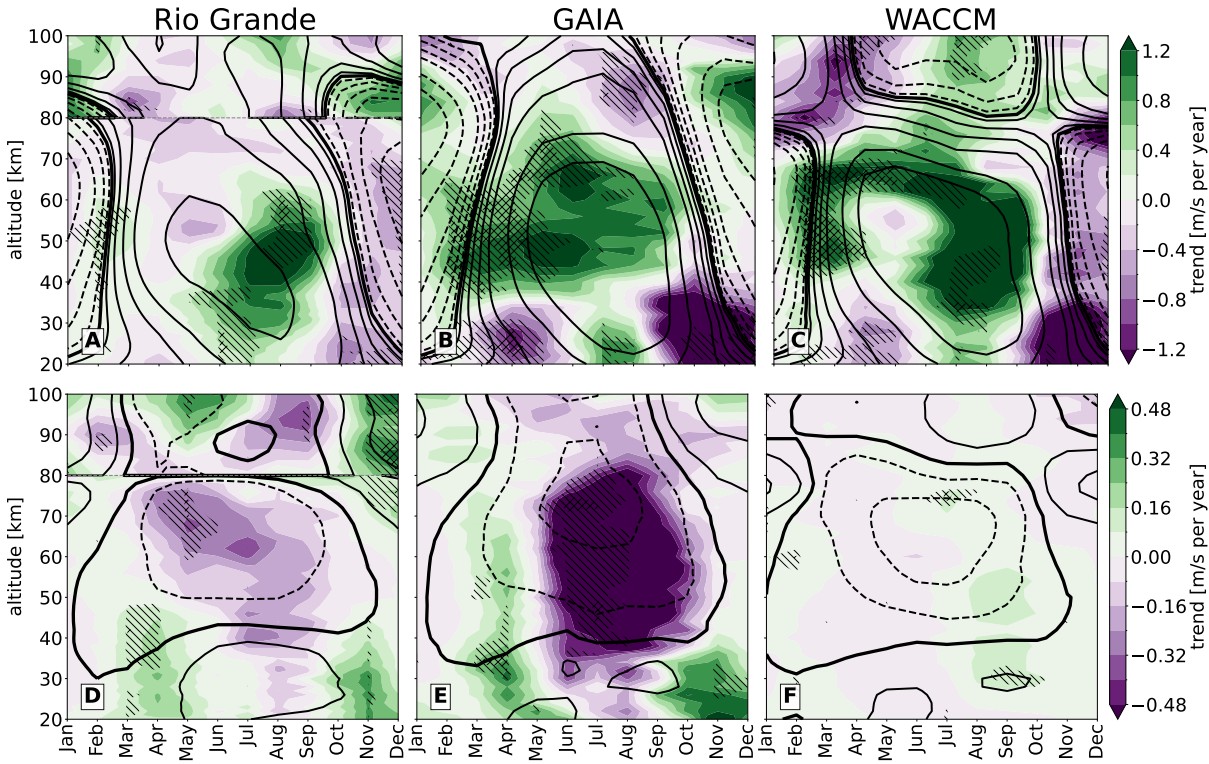

**Figure A4.** Trend comparison (shading) of zonal (upper panels) and meridional (lower panels) monthly mean winds at the locations of Rio Grande between reanalysis/observations (ERA5 and meteor radar measurement for the altitude 20–80 km and 80–100 km, respectively; **A,D**), GAIA (**B,E**), and SD-WACCM-X (**C,F**). The comparison is for the common period 2008–2017. Contours represent wind climatology with the following levels: $0, \pm2, \pm6, \pm12, \pm20, \pm40, \pm60$ m/s. Hatching \\\\ and //// shows where the p-values of the MK test are $< 0.05$ and $< 0.01$, respectively.





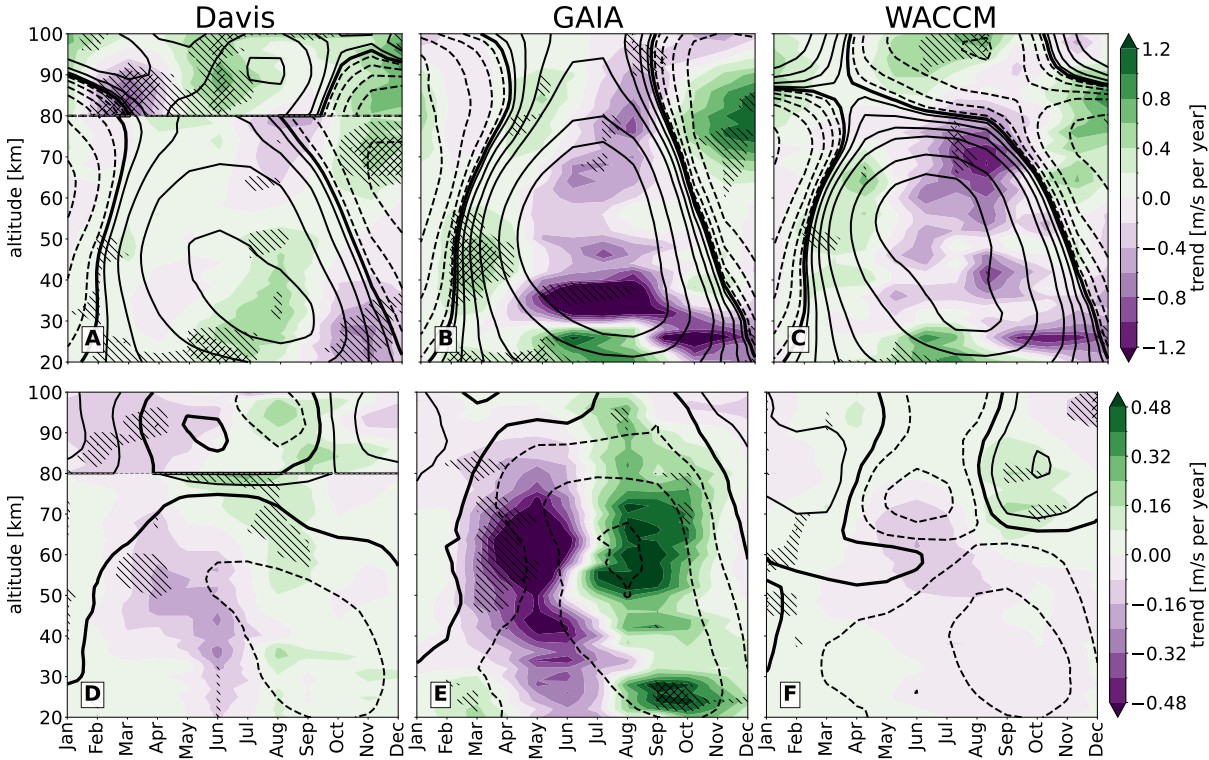

**Figure A5.** Trend comparison (shading) of zonal (upper panels) and meridional (lower panels) monthly mean winds at the locations of Davis between observations (ERA5 and meteor radar measurement for the altitudes 20–80 km and 80–100 km, respectively; **A,D**) and GAIA (**B,E**) and SD-WACCM-X (**C,F**). The comparison is for the common period 2005–2017. Contours represent wind climatology with the following levels: $0, \pm 2, \pm 6, \pm 12, \pm 20, \pm 40, \pm 60$ m/s. Hatching \\\\ and //// shows where the p-values of the MK test are $< 0.05$ and $< 0.01$, respectively.

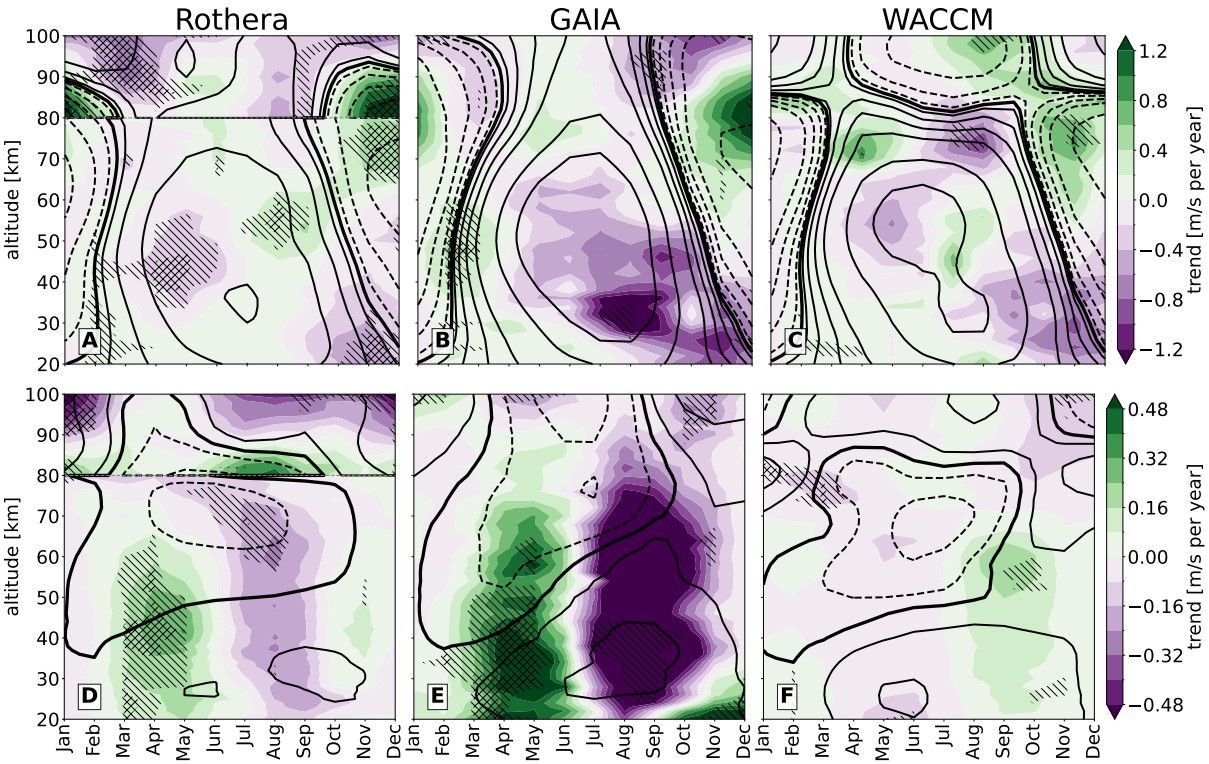

**Figure A6.** Trend comparison (shading) of zonal (upper panels) and meridional (lower panels) monthly mean winds at the locations of Rothera between observations (ERA5 and meteor radar measurement for the altitudes 20–80 km and 80–100 km, respectively; **A,D**) and GAIA (**B,E**) and SD-WACCM-X (**C,F**). The comparison is for the common period 2005–2017. Contours represent wind climatology with the following levels: $0, \pm 2, \pm 6, \pm 12, \pm 20, \pm 40, \pm 60$ m/s. Hatching \\\\ and //// shows where the p-values of the MK test are $< 0.05$ and $< 0.01$, respectively.



*Data availability.* All analysis scripts will be made available on zenodo.com upon acceptance. Similarly, we will make the post-processed simulation data openly available via data.mendeley.com.

*Author contributions.* The conceptual idea of the manuscript was developed by AK, supported by the VACILT project, developed by GS, DP, CJ, and HuL. The data analysis and data reduction were performed by AK, GS, and DP. HuL supported the data analysis and interpretation of GAIA and contributed to the discussion. HLi supported the interpretation of the SD-WACCM-X results and the overall discussion. All authors contributed to the editing and writing of the manuscript.

*Competing interests.* GS and CJ are editors of Annales Geophysicae. The authors declare that there are no further competing interests.

*Acknowledgements.* Gunter Stober is a member of the Oeschger Center for Climate Change Research (OCCR). HL acknowledges support from Japan Society for the Promotion of Science (JSPS Grant JP25K01058, JP22K21345). TMG and the Rothera radar are supported by the NERC NC PRESIENT (UK Polar Research Expertise for Science and Society) programme NE/Y006/78/1.

Operation of the Davis meteor radar was supported by Australian Antarctic Science projects 4025, 4445, and 4637. DJ was supported through the NASA ISFM programs for Heliophysics. SAAMER's operation is supported by NESC assessment TI-517 17-01204. This
research has been supported by the Schweizerischer Nationalfonds zur Förderung der Wissenschaftlichen Forschung (grant no. 200021-200517/1), the Deutsche Forschungsgemeinschaft (grant no. JA 836/47-1), and the International Space Science Institute (ISSI) in Bern (through ISSI International Team project 23-580 – Meteors and Phenomena at the Boundary between Earth's Atmosphere and Outer Space).

The authors would like to thank the relevant working teams for the reanalysis data sets: MERRA2 (Global Modeling and Assimilation Office, GMAO, 2015) and ERA5 (obtained from the Deutsches Klimarechenzentrum (DKRZ), see https://docs.dkrz.de/doc/dataservices/
finding_and_accessing_data/era_data/index.html). For ERA5 processing, the DKRZ resources have been used under project ID bd1022.

Furthermore, we acknowledge developers of python open-source software libraries used for this paper: *pyMannKendall* (Hussain and Mahmud, 2019).



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
