# Peer review of "Ozone recovery effects on mesospheric dynamics in the southern hemisphere"

_EGUsphere, 2025_

## Referee Comment (RC1)

Review of "Ozone recovery effects on mesospheric dynamics in the southern hemisphere"

By A. Kuchar et al.

**Recommendation:** Reject**

This study uses observations made over 2008-2019 at three meteor radar sites located at high latitudes in the Southern Hemisphere, together with reanalysis data and simulations from two high-top models to argue that (1) the transition of the zonal wind from the winter (westerly) regime to the summer (easterly) regime in the stratosphere has trended to earlier dates and (2) that the changes in the zonal wind are reflected in trends in momentum flux and gravity wave potential energy above 80 km obtained from the MR radars and SABER observations, respectively.

I found the conclusion on the earlier zonal wind transition convincing. This is essentially derived from the reanalysis data (MERRA-2 and JRA-25/JRA-55), since the two models used (WACCM-X and GAIA) are constrained by reanalysis in part or all of the stratosphere. The conclusions regarding momentum flux and wave energy density are much more tenuous, as discussed in the specific comments below. The discussion of those measurements glosses over many inconsistencies among the three MR sites to the point that I found the conclusions based on MR data unconvincing.

All in all, the paper is too long and the conclusions from the meteor radars are too weak to warrant publication.

**Specific Comments** (line number)**

- (5) "our results reveal a significant delay in the spring transition": I think you have this backward. The spring transition actually occurs earlier (trends are negative in austral spring). See also comment at (176).
- (16) "our planet's climate": I have not seen any evidence that the MLT plays a role in what is conventionally understood as "climate", that is, the climate of the troposphere. There is also no such claim in the reference cited here, Smith (2012).
- (26) "ZW": I don't see any need to introduce an acronym for zonal wind. It only leads to confusion when the reader forgets what "ZW" stands for. I would suggest eliminating ZW and just writing "zonal wind".
- (44) "reported a decrease in the ZW": Quibbles about the acronym aside, this statement is ambiguous. What does "a decrease" mean? Is it a reduction of the magnitude of the zonal wind, regardless of direction? If so, "weakening" might be a better term to describe the change.
- (62) "the translated effect": I think it would be clearer to write "the impact".

- (95) "parameters ... are given in hourly values": Do you mean that the output frequency is hourly for the variables of interest in the present study? Or are you referring to the model time step?
- (103) "an altitude-extended configuration of ... CESM": More precisely, WACCM-X is in extension into the thermosphere (to about 500 km) of WACCM (the Whole Atmosphere Community Climate Model), which has an upper boundary at about 140 km. Both models use tropospheric physics parameterizations from CAM, the low-top atmospheric component of CESM.
- (105) "upper boundary in the thermosphere": Please state explicitly at what altitude the upper boundary is located.
- (108) "Further details ...": One detail that ought to have been included here is whether this version of WACCM-X is based upon WACCM4 or WACCM6 physics. There are large differences between the two.
- (130) "resulting in large discrepancies in ... ozone": Would such deficiencies (through their role in radiative heating or cooling) impact the zonal wind reanalysis in the mesosphere in MERRA2? I think this deserves at least a comment.
- (135) "SABER data are very sparse": In what sense are SABER data very sparse? As shown by Salby (JAS, 1982), the SABER sampling pattern is equivalent to synoptic sampling with 0.5-day time resolution and horizontal resolution of about 4° in latitude and 6-7 zonal wavenumbers. However, the restriction on high-latitude coverage (poleward of ~53°) due to the 60-day yaw maneuvers is a much more severe problem for present purposes. Is that what you mean?
- (175) "common negative trend in the stratosphere and mesosphere": If you are referring to the strong (and statistically significant) negative trends in zonal wind seen in September through November in the upper panels of Fig. 2 (shaded in purple), these trends are reliably present only in the stratosphere. There is a *slight* extension of negative trends into the mesosphere, to about 60 km, but those trends are not statistically significant above about 40 km.
- (176) "delay in the vortex transition": The significant negative trends in zonal wind occur in austral spring, when the winds in the stratosphere and lower mesosphere are weakening but still westerly. That being the case, a negative trend at this time implies an *advance*, rather than a delay of the seasonal transition from westerlies to easterlies.
- (198) "the same level of compatibility": "the same level of consistency" might be clearer.
- (200) "a weaker observational constraint": I am not sure I understand this. Both components of the monthly-mean, large-scale flow (u,v) ought to be in thermal wind balance at these latitudes to a good approximation.
- (210) "The trend estimate at Rio Grande": The difference between Rio Grande and the other two stations, Rothera and Davis is striking, and not just as regards the trends. The zonal momentum

flux itself differs greatly among locations: it is predominantly negative at Rio Grande throughout the range of altitude shown, whereas it is predominantly positive at Rothera and Davis in SH winter (but not during the rest of the year at Davis). Do you have any idea why? Even if you do not, such a striking difference deserves a comment; it is difficult to have any confidence on explanations of the trends in momentum flux if one cannot even explain the differences in the *sign* of the momentum flux itself among these stations.

- (211) "The weakening and strengthening of westward and eastward...": At Davis, I see negative (i.e., westward) momentum flux starting in September (Fig. 3a). The corresponding trend in flux is positive, so this may be interpreted as a weakening. However, at Rothera (Fig. 3c) the momentum flux is always positive (eastward), so the positive trend starting in September represents a strengthening.
- (214) "Similar to Davis ... Rothera": In what way are Rothera and Davis similar? At Rothera, the momentum flux is always positive (eastward), per Fig. 3c, so a positive trend beginning in September strengthens the flux. On the other hand, at Davis (Fig. 3a) the momentum flux is mostly negative (westward) after September, so a positive trend at that time weakens the flux.
- (215) "a reduction of the westward momentum fluxes in the overall spectrum": It is difficult to understand what this means. Are you saying that, with reduced filtering by the winds at lower altitudes, the spectrum-integrated momentum flux has a smaller net westward magnitude?
- (216) "Above 90 km we observe a strengthening of the westward momentum flux": What Figs. 3a and 3c show is a negative (westward) *trend* in the momentum flux above 90 km. The momentum flux itself is *not* westward (negative) anywhere above 90 km at Davis, and it is westward only above ~95 km at Rothera.
- (217) "These features indicate the coupling via GW": The results from the three MR stations shown in Fig. 3 are so disparate that it is difficult to interpret them. Specifically, the momentum flux itself varies substantially with both altitude and season among the three stations—no two stations show a consistent pattern. The momentum flux trends are roughly similar (but of different magnitude) at Davis and Rothera, but they are either negligible or of the opposite sign at Rio Grande. Given the lack of consistency in both the momentum flux distribution and its trend, I find it impossible to draw useful conclusions.
- (219) "the ZW trends in January and February ...": The zonal wind trends (Fig. 2 a-c) in January-February are much weaker than in September-November and are not always significant. That aside, the trends are positive (green shading), which would indicate an *advanced*, not delayed, transition from the summer (easterly) to the winter (westerly) regime.
- (228) "violin plots": What does the ordinate represent in the "violin plots"? Is it a PDF of slope values? If it is, why do we need a violin? (i.e., does the mirroring about the thin line that shows the median and quartiles add any useful information?)
- (228) "allows us to assess"  $\rightarrow$  allows us to conclude

- (231) "long enough ... to emerge a significant trend" → long enough to detect ... a trend
- (232) "time to emerge": I believe "time to emergence" is the usual phrase
- (243) "Fig. 7 ... GW potential energy ... at all three locations: Add "derived from SABER observations" to remind the reader what data was used to estimate the GW potential energy.
- (244) "with limited data coverage": I am not sure what you mean here. The large data gaps are due to the yaw maneuvers, when no observations poleward of about 53°S are made. That is, there is *no* data during this period rather than "limited data coverage".
- (245) "similarly slight positive trends": Where do you see that in Fig. 7? Austral fall and early winter may reasonably be taken to mean the months of March through June. I do not see "similarly slightly positive trends during these months" at Rio Grande compared to Davis and Rothera of the locations shown. That aside, the trends are for the most part not statistically significant.
- (247) "The negative trend starting in November above 80 km ... corresponds with the weakening of the zonal momentum flux indicated in Figs. 3, 4, 5 and 6": I have no idea what you are trying to state here. In Fig. 7, I can see a clear negative trend in GW potential energy (PE) in November at all three locations. In what way does this correspond to "a weakening of the zonal momentum flux indicated in Figs. 3-6"? Momentum flux is a signed quantity, whereas PE is not, so "weakening" is ambiguous, not to mention that the momentum flux trend in Fig. 3 during November is not uniform across locations or altitude. Adding to the confusion, Figs. 4-5 show a positive trend in momentum flux at Davis (in December, at 82 km), and Fig. 6 has nothing to do with trends but is instead a "time to emergence" plot!
- (264) "probably since": "possibly because" would be better. (There are many reasons why simulations might differ: the upper level of the reanalysis constraint is one; the reanalysis used to constrain the model is another; the response of ozone in the polar cap to the phasing out of ODS is yet another.)
- (264) "the simulation constraint goes higher": This would have been clearer as "the simulation is constrained through a higher altitude (~50 km) in WACCM-X than in GAIA (30 km)".
- (284) "trend slopes in SD-WACCM-X indicate a stronger change": What does this refer to? In Fig. 9d-e for the meridional wind, the WACCM-X trends are everywhere smaller than those in GAIA or the Davis MR.
- (302) "and by thermal tides": Why do you include a reference to thermal tides among coupling mechanisms? You have not investigated the behavior of the tides in this paper.
- (303) "diagnostics strengthen the evidence for vertical coupling": While vertical coupling is a reasonable hypothesis, I find the MR data is insufficient to support it. See previous comments on the momentum flux and GW potential energy results. The response of the momentum flux in the mesosphere, shown in Fig. 3, is inconsistent among the stations. In my view, what this study

shows clearly is that there is an earlier transition from the winter (westerly) regime of zonal wind to the (summer) easterly regime in the lower stratosphere, and that this transition is probably linked to the recovery of Antarctic ozone. Furthermore, this conclusion is ultimately derived mainly from the MERRA-2 and JRA-25/55 reanalysis data because the models (WACCM-X and GAIA) are reanalysis-driven in the stratosphere.

(314) "positive trend in zonal momentum flux around 80 km ... in September is robust": How so? If this statement refers to Fig. 3, Davis and Rothera do show positive (eastward) momentum flux trends between 80 and 90 km but Rio Grande does not. Now, it may be possible to argue that the location of Rio Grande, in the lee of the Andes, somehow makes this station different from the two Antarctic locations, but you have not presented such an argument.

---

## Referee Comment (RC2)

Review of the paper entitled: "Ozone recovery effects on mesospheric dynamics in the southern hemisphere" by Ales Kuchar et al. [MS No.: egusphere-2025-2827].

The paper investigates the impact of ozone recovery on the mesosphere and lower thermosphere (MLT) zonal and meridional wind trends using the meteor radar (MR) and SABER observations over mid-, and high-latitude locations in the southern hemisphere. It further compares the observational results with that from the whole atmosphere models to highlight the discrepancies of the models especially SD-WACCM-X. The authors claim that the ozone recovery impacts the gravity waves (GWs) that propagate and affect the mesospheric winds in spring and early summer. They further use the MERRA2 and ERA5 datasets to compare the winds below 80 km.

Although the paper presents results at 20-100 km, combining MERRA2, ERA5 an MR datasets, the results presented are limited and they do not fundamentally support the given information to evaluate the merits that it claims to report. Significant concerns are due to methodology which does not present complete details of analysis of various datasets including how the trends GW momentum flux and potential energies are estimated from the observations and models. I wonder about using the ozone hole area how it is helpful to correlate with the mesospheric winds? The authors define the ozone hole area as the region of ozone values below 220 DU, however, is this the same for all the months? Moreover it is not specified at which the ozone hole area was estimated. The ozone concentration (in terms of column ozone or the mixing ratios) is more helpful here instead of 'area' as the waves should propagate vertically to reach the mesosphere to affect the winds there. In addition, the tides especially the semidiurnal tide, which is sensitive to the ozone changes, is more effective in modulating the MLT winds. However the role of tides and their interaction with GWs was not discussed in this paper. Further PWs, which have more impact on the polar vortex, have been neglected. The paper focussed more on zonal component but not the meridional winds for the pole-to-pole circulation.

The description of figures is too short, and their detailed scientific merits are not presented. Moreover, the paper does not attempt to address the physical processes to interpret the significant results. The paper is not recommended for publication in the present form.

The specific (major & minor) comments/concerns are listed below.

Title: Include 'mid-high latitude' in the title as this study focuses mainly over these regions.

In the title: mesospheric dynamics, but why the figures represent from 20 km? In the title ozone recovery but no ozone variabities (not area) were shown.

**Abstract:**

L1-2 - How was the recovery of Antarctic ozone 'altered' the stratospheric circulation as the number of recent studies explored this particular topic? Expand GAIA, MERRA2, ERA5, SD-WACCM-X.

L3-Specify the period/duration of the observations/simulations used in this study.

L44- 'They'....who?

L43-45 – decrease in meridional and zonal winds....specify north/south & east/west. Include 'radar' after MF.

L42-46 – Were the trends for all the seasons?

L47-48 – Why not the trends in MLT winds can't be compared before and after 1990? Specify why this particular year.

L51- Include lat. & long. coordinates for Syowa station.

L53-54 – Does winter over Syowa station refer to December-March?

L56- Expand WACCM.

L65 - Replace 'stations' with 'observations'.

L71-72- What is the specific reason for selecting these locations with same latitude/longitudes?

L75-76 – What does the 'methodology' refer here and how did these values obtain?

L79 – Expand SABER. 'vector' or 'magnitude'?

L82- What is meant by 'true'?

L96 - What is 'JRA-25/55'?

L132 – Expand 'MLS'.

L135-137- The 12°x30° (lat. x long.) could be a significant concern of analyzing the MLT winds for individual stations as the dynamics particularly the tidal (atleast non-migrating) forcing could have greater impact on driving the winds in this region.

L142- per 'unit' volume?

L142-143- What is the purpose of GW PE to estimate here and how the 'noisy' and the 'fraction of the data' is helpful here to estimate from the SABER?

L147-148- What causes the seasonal variations of the atmospheric density in MLT and how does it impact the GW activity and background wind....need to explain.

L151- Need brief explanation of ASF2D.

L158- Insert references.

L161- Need brief explanation.

L163-164-What is 'corrupted' here?

L170- Include climatological means of winds for zonal and meridional components prior to the trends.

L172-173- How did these winds retrieve as the 'methodology' didn't reflect it? Further with respect to which year the trends were calculated?

L175 – Specify the height region as the colours are indistinguishable.

L176-177- What could be the physical mechanism for negative trend and how do the ozone recovery linked to the wind reversal? Any explanation for positive trends during May-September?

L177-182- Why the zonal winds are compared with ozone hole area instead ozone concentrations? The ozone concentrations must be presented. Not sure how could the ozone hole area (at which height region) affect the zonal winds at various altitudes of the atmosphere. Further it is confusing 'anti-correated' and 'positive correlations' when comparing with Venkateswara Rao et al. (2015).

L182- What is 'opposite' relation here?

L180-181- Figure A3 showing negative correlations, however Figure A2 displays positive correlations. What is the possible mechanisms behind the two opposite correlations at two height regions?

L192-193- 'twice less' – does it mean half? Why the trends in meridional winds (MWs) are half that of zonal winds (ZWs) as no climatological means are presented here? Further how do the trends in ZWs and MWs are comparable to extract the common trend structure?

L194- 'southward'...do you mean further poleward? Clarify.

L194-195- What could be the reason for negative trend at two different heights at two different stations?

L200-What are these 'constraints' – elaborate.

L202-203- I could not see any positive trends over Davis below 90 km in the summer from Figure 2.

L204- What is the reason for increased upwelling?

Figure 2 – What are A-F labels represent in the figure?

L207- I could not find the positive trends (statistically significant) of ZW at 80 km over any station in September.

L208- Does the figure A3 signifies the trends in ZW or simply ZW variation at 82 km for November?

L207-211 – How significant the ZW trends at 80 km estimated by MRs as the meteor counts are minimum at this height?

L209- How was the zonal momentum flux calculated and then the trends in the same? L211-215- I could see the eastward momentum flux is not statistically significant over Rothera especially after September below ~90 km; however it is significant over Davis and what could be the reason despite the two stations being on the same latitude band. L217- The main concern is that no result was showed for the link/coupling between GWs and the ozone recovery. What is the direct impact of ozone (depletion/recovery)on the GWs as well as the MLT winds? Must be specified.

L216-217 and Figures 2 & 3 – I could see the westward trend of GW momentum flux during May-Sept. over Davis from Fig. 3A, however, the trend in ZWs are eastward during this period above 80 km. Same for Rothera above 90 km. Why? Comparing to Figs. 2B and 3B, what could be the reason for negative and positive trends of ZWs below and above 90 km during Nov.-Feb. over RioGrande? Is there any link between GW MF and ZW trends? What is the 'slope' in Fig. 3B?

L219-222 – I could see transition of westward trends to eastward around at 90 km over all the stations during Nov.-Feb.

L219-222- I wonder what is the transition refer here with Jan.-Feb. being the summer in SH? Further what are 'weaker' and 'stronger' refer to as the eastward MF is more significant over Davis than that over Rothera during Jan. – Feb.? Also why the westward MF above 90 km is more persistent all the year over Rothera than Davis being both the stations located at the same latitude band?

L223-229- Explain how these values are calculated and include appropriate references. Further is this valid over other two stations as no results are presented?

Figure 5 – This figure needs more explanation with additional details to understand the importance of these comparisons.

L230-236 — Since the period 2005-2021 is longer than a 11-year solar cycle, this time series is enough to derive the trends. If that is the case, what is the need of Fig. 6 and not enough details are explained to understand this figure and its relevance to Fig. 4. Further fewer years i.e. 8 years are inadequate to retrieve the trends.

L239-240 & L241-242- Which correct...above 200 km or 500 km? Further what are the GW parameters like vertical wavelength and periods as both ground based and space borne measurements have different horizontal and vertical resolutions and periods. L243-What is the data source and how was the GW potential energy (PE) calculated and trends in it? Should be explained. Why it was limited upto 90 km only?

L243-245- Did you omit the time series of MR GW PE as same as shown in Fig. 7 for better comparisons?

L245-246- I could see the negative trends instead positive.

L247- What does weakening winds mean...eastward or westward?

L249- Why does the negative trends stronger at Davis than at Rothera despite at the same latitude?

L251- What are net momentum fluxes? Explain.

 $L255 - E_{pot}$  and not Epot.

L261-262- Here and wherever applicable – Do MR observations have continuous measurements or any data gaps exist? No where this was mentioned. Further data gaps could be serious concern when compare with other datasets like WACCM.

Figure 8 – It is well known that the WACCM doesn't reproduce the winter westerlies (eastward winds) over mid-high latitudes in the upper MLT. I am wondering how did the authors obtain positive trends above ~85 km in the model. Further as trend means changes on a time scale longer than a solar cycle (11-year), the data shown here are inadequate. How were the trends in winds from the models calculated?

Figures 8A & D were already shown and repeated from Figure 2. Same for Figures 9 and 10.

Figure 8F - As the WACCM reproduces winter southward and northwards winds below and above ~90 km, why the trends shown in this figure are insignificant and do not agree with the observations in the upper MLT?

L274- If the focus of this study was the mesosphere, why the trends shown for all layers from 20 km? This applies to all figures and lines.

L275- Specify here what is meant by beginning and end of summer as I could not find negative trends in the beginning and positive trends in the end of the summer?

L279- Remove '.' I wonder why most of the trends are statistically insignificant? Further why the trends are positive in Nov./Dec. as it contradicts with the westward zonal flow below  $\sim$ 90 km in winter including Oct. – Mar.?

L281-282- I could not see the WACCM reproduction of the long-term changes in zonal winds.

L283-285- I trust the focus of this study is of the mesosphere and not the below layers, and I could not see any negative and positive trends in the meridional winds during the said period at the heights of meteor radar observations.

L289- I wonder how the trends are positive if the zonal flow is westward during Nov. – Feb. below 90 km?

290- I could see the opposite trends during these months below 80 km.

L291 – Again I still believe the focus of this study is mesosphere and not stratosphere.

L296- I wonder no result has been presented on the ozone recovery in terms of concentration/vmr and this led to misinterpretation on the wave dynamics and in turn the MLT winds over Antarctic.

L296-297- Is this 'evident' from any result shown here?

L301 – In this case, no results on PWs and tides were shown. It is very important to focus on these wave phenomena especially over the polar latitudes in the context of ozone variability.

L305 – Which 'analysis'?

Figure A2 – What is the significance of this figure at 30 km? Why the blue lines (ozone hole area) are different from those in Figure A3?

---

## Author Comment (AC1)

**Replies to reviews for:**
**Ozone recovery effects on mesospheric dynamics in the southern hemisphere**

ANGEO, Ales Kuchar et al.

**1 Reply to editor**

Dear editor,

Thank you very much for helping in the publishing process of this paper!

We have revised the manuscript by addressing the comments of the two anonymous reviewers. Main changes are:

5    – Following the suggestion of Reviewer #2 we have changed the title of the paper

– We have clarified that Rio Grande is a special case because it is located at a very pronounced gravity wave hotspot where secondary gravity waves are more dominant than at the other two stations. Therefore it is difficult to compare observations at Rio Grande with the other two stations.

– The mechanism of gravity wave filtering by the background winds has been explained in more detail and the discussion
10    of physical processes has been strengthened

– Several inconsistencies and inaccuracies of the trend discussion have been resolved

– In order to streamline the manuscript, details of the trend determination methodology have been moved to the Appendix

– We have added in the Appendix an investigation to make sure that an issue regarding the SD-WACCM-X v2.1 simulations does not affect our results

15    We hope that the revised manuscript can be published in ANGEO.

Sincerely,
Ales Kuchar and coaouthors

**2 Reply to Ref #1**

Dear anonymous reviewer #1,

We appreciate your very helpful comments! We revised the manuscript accordingly, such that we hope it can then be published in ANGEO. Please find our point-by-point answers to the comments below.

We would like to also openly inform about an issue pertaining to the use of existing SD-WACCM-X v2.1 simulations (https://doi.org/10.26024/5b58-nc53) which has recently been brought to our attention. In the course of performing the sim-

25 ulations, occasional instabilities in the model necessitated restarting from an initial condition file. This inadvertently led to resetting $CO_2$ concentrations to year 2000 conditions in the atmosphere. This is illustrated in Fig. R1, which shows the global average $CO_2$ at the surface (solid lines) and at $10^{-2}$ hPa (dashed lines) in the SD-WACCM-X v2.1 simulations (red) and a corresponding historical free run (blue). Occasional jumps in $CO_2$ concentrations are evident in the SD-WACCM-X v2.1 simulation at $10^{-2}$ hPa that correspond to times when the model was reinitialized. These do not occur at the surface where $CO_2$

30 concentration is specified or in the historical free-run which was not reinitialized at any point in the simulations.

[Figure]

**Figure R1.** Global average $CO_2$ concentration in SD-WACCM-X v2.1 (red) and a corresponding historical free-run (blue) at the surface (solid) and $10^{-2}$ hPa (dashed). The periodic jumps in $CO_2$ concentration at $10^{-2}$ hPa are due to reinitialization of the model following the occurrence of numerical instabilities.

We dedicated an subsection in the Appendix to discuss this issue. Time series of zonal and meridional wind at 80 km at all stations therein do not indicate that these $CO_2$ jumps impacted the winds locally and correspondingly the results in Section 4.3.

Users of the SD-WACCM-X v2.1 simulations should be aware of the inaccurate $CO_2$ concentrations at higher altitudes. An updated SD-WACCM-X simulation from 1980-present will be performed and released to the community following complete

35  understanding of this issue. We thank McArthur Jones Jr. and John Emmert (Naval Research Laboratory) for bringing this issue to our attention.

    Best wishes

    Ales Kuchar et al

40  ## 2.1  Major comments

    ## 2.2  Specific comments

    (5) "our results reveal a significant delay in the spring transition": I think you have this backward. The spring transition actually occurs earlier (trends are negative in austral spring). See also comment at (176).

    The sentence in the abstract has been reworded as follows:

45  "Coinciding with the earlier breakdown of the stratospheric vortex, we observe a significant weakening of the upper mesospheric easterlies (around 82 km), i.e. an opposite trend."

    (16) "our planet's climate": I have not seen any evidence that the MLT plays a role in what is conventionally understood as "climate", that is, the climate of the troposphere. There is also no such claim in the reference cited here, Smith (2012).

50  We deleted "our planet's climate and". We also changed the reference to Smith (2012).

    (26) "ZW": I don't see any need to introduce an acronym for zonal wind. It only leads to confusion when the reader forgets what "ZW" stands for. I would suggest eliminating ZW and just writing "zonal wind".

    We eliminated the abbreviation accordingly.

55

    (44) "reported a decrease in the ZW": Quibbles about the acronym aside, this statement is ambiguous. What does "a decrease" mean? Is it a reduction of the magnitude of the zonal wind, regardless of direction? If so, "weakening" might be a better term to describe the change.

    We changed to "weakening".

60

    (62) "the translated effect": I think it would be clearer to write "the impact".

    We changed accordingly.

    (95) "parameters . . . are given in hourly values": Do you mean that the output frequency is hourly for the variables of interest

65  in the present study? Or are you referring to the model time step?

We changed the sentence to "The output frequency of simulated atmospheric parameters (e.g., wind, temperature) is on hourly basis".

(103) "an altitude-extended configuration of ... CESM": More precisely, WACCM-X is in extension into the thermosphere (to about 500 km) of WACCM (the Whole Atmosphere Community Climate Model), which has an upper boundary at about 140 km. Both models use tropospheric physics parameterizations from CAM, the low-top atmospheric component of CESM.
We modified the description accordingly.

(105) "upper boundary in the thermosphere": Please state explicitly at what altitude the upper boundary is located.
We added "(500-700 km altitude depending on solar activity).".

(108) "Further details ...": One detail that ought to have been included here is whether this version of WACCM-X is based upon WACCM4 or WACCM6 physics. There are large differences between the two.
We now noted in the manuscript that this version of WACCM-X is based on the CAM4 physics.

(130) "resulting in large discrepancies in ... ozone": Would such deficiencies (through their role in radiative heating or cooling) impact the zonal wind reanalysis in the mesosphere in MERRA2? I think this deserves at least a comment.
We noted in the revised manuscript that these discrepancies could potentially impact the zonal wind in the mesosphere in MERRA2. However, the compatibility between MERRA2 and meteor radars provides confidence that zonal winds are sufficiently constrained by temperature observations.

(135) "SABER data are very sparse": In what sense are SABER data very sparse? As shown by Salby (JAS, 1982), the SABER sampling pattern is equivalent to synoptic sampling with 0.5-day time resolution and horizontal resolution of about 4° in latitude and 6-7 zonal wavenumbers. However, the restriction on high-latitude coverage (poleward of 53°) due to the 60-day yaw maneuvers is a much more severe problem for present purposes. Is that what you mean?
We revised to "Given the strong intermittency of gravity waves, a sufficient number of data points is needed for obtaining robust averages (see also Ern et al. (2022)). Therefore, we averaged..."

(175) "common negative trend in the stratosphere and mesosphere": If you are referring to the strong (and statistically significant) negative trends in zonal wind seen in September through November in the upper panels of Fig. 2 (shaded in purple), these trends are reliably present only in the stratosphere. There is a slight extension of negative trends into the mesosphere, to about 60 km, but those trends are not statistically significant above about 40 km.
We added "while statistically significant below 40 km".

100     (176) "delay in the vortex transition": The significant negative trends in zonal wind occur in austral spring, when the winds in the stratosphere and lower mesosphere are weakening but still westerly. That being the case, a negative trend at this time implies an advance, rather than a delay of the seasonal transition from westerlies to easterlies.

    We revised this inconsistency to: "causing an advance of the vortex transition from westerlies to easterlies". We also revised it in l306.

105     (198) "the same level of compatibility": "the same level of consistency" might be clearer.

    We changed accordingly.

    (200) "a weaker observational constraint": I am not sure I understand this. Both components of the monthly-mean, large-scale flow (u,v) ought to be in thermal wind balance at these latitudes to a good approximation.

110     To avoid confusion we deleted the second part of this sentence ("...due to ...").

    (210) "The trend estimate at Rio Grande": The difference between Rio Grande and the other two stations, Rothera and Davis is striking, and not just as regards the trends. The zonal momentum flux itself differs greatly among locations: it is predominantly negative at Rio Grande throughout the range of altitude shown, whereas it is predominantly positive at Rothera

115 and Davis in SH winter (but not during the rest of the year at Davis). Do you have any idea why? Even if you do not, such a striking difference deserves a comment; it is difficult to have any confidence on explanations of the trends in momentum flux if one cannot even explain the differences in the sign of the momentum flux itself among these stations.

    At a given location, the sign of net gravity wave momentum flux depends on: (1) the local sources of gravity waves in the troposphere and lower stratosphere, (2) the filtering of this upward propagating spectrum of "primary" gravity waves by the

120 vertical profile of the horizontal wind, (3) gravity waves that may be refracted horizontally into the observed air volume, and, (4) the spectrum of secondary gravity waves which is excited when the "primary" gravity waves break in the stratosphere and mesosphere (which should be most relevant at the Southern Andes hotspot of orographic GWs, i.e. Rio Grande).

    These factors may differ between different stations and could contribute to the observed different signs of the momentum flux at different stations. This means it is not too surprising that the net gravity wave momentum flux at different stations

125 can have different sign. For example, de Wit et al. (2017) found an unexpected sign of gravity wave momentum flux at Rio Grande in the lee of the Andes and attributed it to the generation of secondary gravity waves in the stratosphere. Such detailed discussion about the sign of the momentum fluxes at the different stations, however, is beyond the scope of our current work.

    Of course, also trends in the momentum fluxes can have different reasons. For Davis and Rothera, however, it seems that the trends are caused by the simple mechanism of wind filtering of upward propagating waves: strong westward trend of zonal wind

130 is observed in September to November in the stratosphere. Therefore gravity waves of westward phase speed should be more strongly filtered out, and, correspondingly, near the mesopause (below $90\,\mathrm{km}$) trends of net gravity wave momentum fluxes should be positive as eastward propagating gravity waves should become more dominant in the overall spectrum of gravity waves. This mechanism will act independent of the sign of net gravity wave momentum flux. For Rio Grande this simple explanation does not seem to hold. One reason could be that zonal wind trends in the stratosphere are weaker than at Davis

135    and Rothera (see Fig. 2). However, also other factors could contribute. For example, the wind filtering of waves propagating upward from the troposphere and lower stratosphere could be masked by excitation of secondary gravity waves which was seen by de Wit et al. (2017) at the same location.

    This will be pointed out more clearly in the revised manuscript.

140    (211) "The weakening and strengthening of westward and eastward...": At Davis, I see negative (i.e., westward) momentum flux starting in September (Fig. 3a). The corresponding trend in flux is positive, so this may be interpreted as a weakening. However, at Rothera (Fig. 3c) the momentum flux is always positive (eastward), so the positive trend starting in September represents a strengthening.

    Yes, that's what we actually state: "The weakening and strengthening of westward and eastward zonal momentum flux at
145    Davis (Fig. 3**A**) and Rothera (Fig. 3**C**), respectively, starts in October below 90 km..."

    As stated above, also trends in momentum fluxes can have different reasons. Therefore the trend at Davis starting in September could just be a general weakening, and the trend at Rothera a general strengthening of momentum fluxes.

    (214) "Similar to Davis . . . Rothera": In what way are Rothera and Davis similar? At Rothera, the momentum flux is always
150    positive (eastward), per Fig. 3c, so a positive trend beginning in September strengthens the flux. On the other hand, at Davis (Fig. 3a) the momentum flux is mostly negative (westward) after September, so a positive trend at that time weakens the flux.

    The results from SABER indicate a weakening of gravity wave potential energy at both locations. This supports our explanation of the strengthening wind filtering of just the westward propagating gravity waves, which would result in a general weakening of absolute momentum fluxes or potential energy.

155

    (215) "a reduction of the westward momentum fluxes in the overall spectrum": It is difficult to understand what this means. Are you saying that, with reduced filtering by the winds at lower altitudes, the spectrum-integrated momentum flux has a smaller net westward magnitude?

    Yes, you are correct. By the strengthening trend of wind filtering of westward propagating gravity waves the spectrum of
160    gravity waves is shifted more towards the eastward phase speeds, i.e., the integrated magnitude of westward momentum flux is reduced.

    This will be clarified in the revised manuscript.

    (216) "Above 90 km we observe a strengthening of the westward momentum flux": What Figs. 3a and 3c show is a negative
165    (westward) trend in the momentum flux above 90 km. The momentum flux itself is not westward (negative) anywhere above 90 km at Davis, and it is westward only above 95 km at Rothera.

    Reviewer 1 is correct, just a westward trend in momentum flux is observed. We revised accordingly.

(217) "These features indicate the coupling via GW": The results from the three MR stations shown in Fig. 3 are so disparate that it is difficult to interpret them. Specifically, the momentum flux itself varies substantially with both altitude and season among the three stations—no two stations show a consistent pattern. The momentum flux trends are roughly similar (but of different magnitude) at Davis and Rothera, but they are either negligible or of the opposite sign at Rio Grande. Given the lack of consistency in both the momentum flux distribution and its trend, I find it impossible to draw useful conclusions.

Of course, it is interesting that the trends in Rio Grande are different from those at Davis and Rothera. However, given the fact that Tierra del Fuego was previously identified as a special case by de Wit et al. (2017), we still think that the trend results for Davis and Rothera are consistent and can be explained in a simple way by wind filtering of vertically propagating gravity waves, which would be a mechanism of vertical coupling between different layers of the atmosphere.

(219) "the ZW trends in January and February ...": The zonal wind trends (Fig. 2 a-c) in January-February are much weaker than in September-November and are not always significant. That aside, the trends are positive (green shading), which would indicate an advanced, not delayed, transition from the summer (easterly) to the winter (westerly) regime.

(228) "violin plots": What does the ordinate represent in the "violin plots"? Is it a PDF of slope values? If it is, why do we need a violin? (i.e., does the mirroring about the thin line that shows the median and quartiles add any useful information?)

In the revised manuscript, we changed Fig. 5**B** so that it does not mirror the zonal momentum flux distribution. We also revised the figure description. Accordingly, we revised Fig. 6.

(228) "allows us to assess" -> "allows us to conclude"
We revised it accordingly.

(231) "long enough ... to emerge a significant trend" -> "long enough to detect ... a trend"
We revised it accordingly.

(232) "time to emerge": I believe "time to emergence" is the usual phrase
We revised it accordingly.

(243) "Fig. 7 ... GW potential energy ... at all three locations: Add "derived from SABER observations" to remind the reader what data was used to estimate the GW potential energy.
"derived from SABER observations" was added.

(244) "with limited data coverage": I am not sure what you mean here. The large data gaps are due to the yaw maneuvers, when no observations poleward of about 53°S are made. That is, there is no data during this period rather than "limited data coverage".

We are sorry for this unspecific statement. Strictly speaking, there are some data in the white spaces. The times when the TIMED satellite performs yaw maneuvers gradually shifted somewhat to earlier dates over the course of the SABER mission. Therefore in some of the white spaces data exist, but only for a few years.

(245) "similarly slight positive trends": Where do you see that in Fig. 7? Austral fall and early winter may reasonably be taken to mean the months of March through June. I do not see "similarly slightly positive trends during these months" at Rio Grande compared to Davis and Rothera of the locations shown. That aside, the trends are for the most part not statistically significant.

Reviewer 1 is correct, the slightly positive trends in March are only seen at Davis and Rothera, and in May at Rio Grande. We revised accordingly.

(247) "The negative trend starting in November above 80 km ... corresponds with the weakening of the zonal momentum flux indicated in Figs. 3, 4, 5 and 6": I have no idea what you are trying to state here. In Fig. 7, I can see a clear negative trend in GW potential energy (PE) in November at all three locations. In what way does this correspond to "a weakening of the zonal momentum flux indicated in Figs. 3-6"? Momentum flux is a signed quantity, whereas PE is not, so "weakening" is ambiguous, not to mention that the momentum flux trend in Fig. 3 during November is not uniform across locations or altitude. Adding to the confusion, Figs. 4-5 show a positive trend in momentum flux at Davis (in December, at 82 km), and Fig. 6 has nothing to do with trends but is instead a "time to emergence" plot!

Indeed, it should be Figs. 3–5; Fig. 6 does not fit here. What we meant is that a weakening of potential energy suggests that the magnitude of momentum flux (i.e. absolute momentum flux) should also be weakening. This is discussed in more detail in our reply to (211) and (214) where we argue that in November at Davis and Rothera predominantly westward propagating gravity waves encounter wind filtering in the stratosphere, and the westward trend of the zonal wind will strengthen this filtering, which should result in more eastward net momentum fluxes, but reduced absolute momentum fluxes and reduced potential energy.

(264) "probably since": "possibly because" would be better. (There are many reasons why simulations might differ: the upper level of the reanalysis constraint is one; the reanalysis used to constrain the model is another; the response of ozone in the polar cap to the phasing out of ODS is yet another.)

We modified as recommended.

(264) "the simulation constraint goes higher": This would have been clearer as "the simulation is constrained through a higher altitude ( 50 km) in WACCM-X than in GAIA (30 km)".

We revised accordingly.

(284) "trend slopes in SD-WACCM-X indicate a stronger change": What does this refer to? In Fig. 9d-e for the meridional wind, the WACCM-X trends are everywhere smaller than those in GAIA or the Davis MR.

What we meant is that both positive and negative trends for WACCM have somewhat larger magnitudes at Davis than at the other two stations. However, the magnitudes are still much smaller than for the radar observations/MERRA2 and therefore not very realistic. We revised part of this sentence accordingly.

(302) "and by thermal tides": Why do you include a reference to thermal tides among coupling mechanisms? You have not investigated the behavior of the tides in this paper.

(303) "diagnostics strengthen the evidence for vertical coupling": While vertical coupling is a reasonable hypothesis, I find the MR data is insufficient to support it. See previous comments on the momentum flux and GW potential energy results. The response of the momentum flux in the mesosphere, shown in Fig. 3, is inconsistent among the stations. In my view, what this study shows clearly is that there is an earlier transition from the winter (westerly) regime of zonal wind to the (summer) easterly regime in the lower stratosphere, and that this transition is probably linked to the recovery of Antarctic ozone. Furthermore, this conclusion is ultimately derived mainly from the MERRA-2 and JRA-25/55 reanalysis data because the models (WACCM-X and GAIA) are reanalysis-driven in the stratosphere.

We are sorry for mixing a general introductory statement with "our findings". Further we will down-tone the statement about our findings. Lines 301+302+303 will be reworded as follows: "One mechanism for coupling between the stratosphere and the MLT region is wave coupling by PWs, GWs of various scales, and by thermal tides. In our work we provide additional evidence for the role of coupling by GWs using both ..."

(316) "positive trend in zonal momentum flux around 80 km ... in September is robust": How so? If this statement refers to Fig. 3, Davis and Rothera do show positive (eastward) momentum flux trends between 80 and 90 km but Rio Grande does not. Now, it may be possible to argue that the location of Rio Grande, in the lee of the Andes, somehow makes this station different from the two Antarctic locations, but you have not presented such an argument.

Indeed, the location in the lee of the Andes (a hotspot of strong orographic GWs; Ern et al., 2018). Rio Grande differs from the other stations because the MLT region there seems to be dominated by secondary gravity waves, and not by upward propagating primary gravity waves. This has been investigated in more detail in de Wit et al. (2017). This will be stated more clearly in the revised manuscript.

**3   Reply to Ref. #2**

Dear anonymous reviewer #2,
thank you very much for your valuable comments. Please find our point-by-point answers (in blue) to your comments below. Furthermore, we changed the title according to your own suggestion.

270    We would like to also openly inform about an issue pertaining to the use of existing SD-WACCM-X v2.1 simulations (https://doi.org/10.26024/5b58-nc53) which has recently been brought to our attention. In the course of performing the simulations, occasional instabilities in the model necessitated restarting from an initial condition file. This inadvertently led to resetting $CO_2$ concentrations to year 2000 conditions in the atmosphere. This is illustrated in Fig. R1, which shows the global average $CO_2$ at the surface (solid lines) and at $10^{-2}$ hPa (dashed lines) in the SD-WACCM-X v2.1 simulations (red) and a
275    corresponding historical free run (blue). Occasional jumps in $CO_2$ concentrations are evident in the SD-WACCM-X v2.1 simulation at $10^{-2}$ hPa that correspond to times when the model was reinitialized. These do not occur at the surface where $CO_2$ concentration is specified or in the historical free-run which was not reinitialized at any point in the simulations.

We dedicated an subsection in the Appendix to discuss this issue. Time series of zonal and meridional wind at 80 km at all stations therein do not indicate that these $CO_2$ jumps impacted the winds locally and correspondingly the results in Section 4.3.

280    Users of the SD-WACCM-X v2.1 simulations should be aware of the inaccurate $CO_2$ concentrations at higher altitudes. An updated SD-WACCM-X simulation from 1980-present will be performed and released to the community following complete understanding of this issue. We thank McArthur Jones Jr. and John Emmert (Naval Research Laboratory) for bringing this issue to our attention.

Best wishes

285  Ales Kuchar et al

**3.1  Major comments**

**3.2  Specific comments**

290  Title: Include 'mid-high latitude' in the title as this study focuses mainly over these regions.

In the title: mesospheric dynamics, but why the figures represent from 20 km? In the title ozone recovery but no ozone variabilies (not area) were shown.

We changed the title to "Ozone recovery effects on stratospheric and lower-mesospheric mid-high latitude dynamics in the southern hemisphere".

295

Abstract: L1-2 - How was the recovery of Antarctic ozone 'altered' the stratospheric circulation as the number of recent studies explored this particular topic? Expand GAIA, MERRA2, ERA5, SD-WACCM-X.

We revised the opening to be specific and literature-consistent: ""Antarctic ozone recovery has contributed to springtime weakening and an earlier breakdown of the Southern Hemisphere stratospheric polar vortex."" All acronyms are expanded at
300  first use

L3-Specify the period/duration of the observations/simulations used in this study.

We added explicit periods of the observations/simulations used in this study.

L44- 'They'….who?

305    We replaced "They" with "Both studies".

L43-45 – decrease in meridional and zonal winds….specify north/south & east/west. Include 'radar' after MF.

We should have been more specific. We changed this to "Trend analyses of the northward meridional wind until 1986 showed a decrease in southward winds during winter, leading to a decrease of annual mean positive (northward) winds (Portnyagin

310    et al., 1993; Bremer et al., 1997). These studies also reported a weakening of the eastward zonal wind in summer and winter (Portnyagin et al., 1993) and annual mean (Bremer et al., 1997),…"

L42-46 – Were the trends for all the seasons?

We changed this. Meridional winds had the trends in winter, zonal winds in both summer and winter

315

L47-48 – Why not the trends in MLT winds can't be compared before and after 1990? Specify why this particular year.

We meant that the trends before/after differ. Actually, regarding the timing, different datasets show somethat different break years. We modified this part to be more specific: "...confirmed by Merzlyakov et al. (2009) who added MF radar observations from Mawson (68°S, 63°E) since 1984. They showed, however, that (after a data gap 1991 - 1993) there was a step change

320    towards more westward winds in summer, and after that a change towards a positive summer zonal wind trend beginning in 1994. Changes of annual mean zonal wind trends from negative to positive have also been observed at northern hemisphere latitudes (Portnyagin et al., 2006), but around 1985 or 1993, depending on the station. One may conclude that trends before the 1990s are different from those detected from later observations."

325    L51- Include lat. & long. coordinates for Syowa station.

We added its coordinates.

L53-54 – Does winter over Syowa station refer to December-March?

We omit "in winter" in this sentence.

330

L56- Expand WACCM.

We added "the specified dynamics version of".

L65 – Replace 'stations' with 'observations'.

335    We replaced 'stations' with 'observations'.

L71-72- What is the specific reason for selecting these locations with same latitude/longitudes?

There is a limited number of stations in and around Antarctica covering a longer time span. Having 3 stations with similar latitudes or longitudes provides additional information on whether or not there are latitudinal /longitudinal differences.

L75-76 – What does the 'methodology' refer here and how did these values obtain?

We replaced it with "using the adaptive spectral filter (ASF2D; see Section 3.1)"

L79 – Expand SABER. 'vector' or 'magnitude'?

We revised the sentenced towards: "..., i.e. SABER provides only GW potential energies and absolute GW momentum fluxes."

L82- What is meant by 'true'?

We added the following footnote: "'True' horizontal wavelength here means the correct physical wavelength of a GW measured perpendicular to its wave fronts. This wavelength differs from the "apparent" wavelengths along the measurement track that can only be detected by single-track satellite instruments, or the wavelengths parallel to the line of sight that are relevant for the instrument sensitivity to GWs."

L96 – What is 'JRA-25/55'?

We introduced "JRA" as an abbreviation in the manuscript.

L132 – Expand 'MLS'.

We replaced "MLS" with "Microwave Limb Sounder (MLS)".

L135-137- The 12°x30° (lat. x long.) could be a significant concern of analyzing the MLT winds for individual stations as the dynamics particularly the tidal (at least nonmigrating) forcing could have greater impact on driving the winds in this region.

Areas of this size are needed to have sufficient statistics for obtaining robust averages from the SABER GW data given the fact that GWs are very intermittent (e.g. Ern et al., 2022).

L142- per 'unit' volume?

yes, "per unit volume"; was corrected

L142-143- What is the purpose of GW PE to estimate here and how the 'noisy' and the 'fraction of the data' is helpful here to estimate from the SABER?

As described in Ern et al. (2018), GW potential energies can be calculated for each SABER altitude profile. However, the statistics for SABER GW momentum fluxes is much worse because two conditions have to be fulfilled: (1) short enough

along-track sampling distance between the two profiles of an along-track pair of profiles. This condition is only fulfilled for every second pair of SABER profiles. (2) The same GW has to be observed in an along-track pair of SABER profiles. This is made sure by a vertical wavelength matching criterion. Usually only about 60of the remaining SABER along-track pairs after criterion (1) fulfill this second criterion.

In the text we added the following: "As detailed in Ern et al. (2018) the statistics for the SABER GW potential energies is about four times better than for GW momentum fluxes. This makes averages of GW potential energies more robust and less noisy."

L147-148- What causes the seasonal variations of the atmospheric density in MLT and how does it impact the GW activity and background wind. . . .need to explain.

We added the reference Strelnikova et al. (2021) in the manuscript where the effect of atmospheric density is discussed. This publication documents that there are seasonal variations and we take them into account.

L151- Need brief explanation of ASF2D.

We added the description of ASF2D in this subsection.

L158- Insert references.

The sentence refers to Wilhelm et al (2019). To make this clear, we slightly modified it to "Based on the decomposed time series, they analyzed long-term changes..."

L161- Need brief explanation.

We added the following "is an unbiased estimator of the true slope in simple linear regression".

L163-164-What is 'corrupted' here?

It means that the Theil-Sen estimate can tolerate arbitrary corruption of up to 29.3% of input data points (outliers) without degradation of its accuracy. Therefore, we added "outliers" in the manuscript.

L170- Include climatological means of winds for zonal and meridional components prior to the trends.

Solid (positive values) and dashed (negative values) contours represent wind climatology and accompany the trend estimates in our results. We mention this at the beginning of Section 4.

L172-173- How did these winds retrieve as the 'methodology' didn't reflect it? Further with respect to which year the trends were calculated?

405    We added that "monthly mean winds are derived from ASF2D mean-wind components (MR) and from gridded model/reanalysis winds interpolated to the station region".

L175 – Specify the height region as the colours are indistinguishable.

We added the following: "at the height region 20 – 60 km, but with weaker magnitude and not significant above 30 – 35 km."

410

L176-177- What could be the physical mechanism for negative trend and how do the ozone recovery linked to the wind reversal? Any explanation for positive trends during May-September?

We discussed the physical mechanism below (L176-177) in the text.

The positive trends during May-September when westerly occurs in SH may be explained by enhanced meridional temper-

415    ature gradient through radiative cooling in higher latitudes associated with enhanced ozone concentrations. A similar pattern was reported in studies (Banerjee et al., 2020; Song and Song, 2024) already mentioned in the manuscript. Solomon et al. (2017) also reveal observed and modeled patterns of enhanced stratospheric cooling during the recovery period which may lead to positive trends in zonal wind. We now discuss this in the manuscript.

420    L177-182- Why the zonal winds are compared with ozone hole area instead ozone concentrations? The ozone concentrations must be presented. Not sure how could the ozone hole area (at which height region) affect the zonal winds at various altitudes of the atmosphere. Further it is confusing 'anti-correated' and 'positive correlations' when comparing with Venkateswara Rao et al. (2015).

The time series of ozone hole area represents the region of depleted ozone known as the "ozone hole". The area of the ozone

425    hole is determined from a map of total column ozone. It is calculated from the area on the Earth that is enclosed by a line with a constant value of 220 Dobson Units (DU). The value of 220 DU is chosen since total ozone values of less than 220 DU were not found in the historic observations over Antarctica prior to 1979. It is considered as a primary metric to monitor ozone-depleting substances and track recovery (e.g. Newman et al., 2006). This metric also avoids an arbitrary choice of altitude or region of ozone concentrations.Venkateswara Rao et al. (2015) shows that their findings are robust across various measures of ozone

430    loss.

We revised the sentence to "Similarly to Venkateswara Rao et al. (2015), we present evidence that the stratospheric and mesospheric wind strengths are strongly affected by the perturbations induced by the Antarctic ozone hole."

L182- What is 'opposite' relation here?

435    We revised the sentences to "...reveals even higher but negative correlations ($r \leq -0.63$). We attribute the opposite relation to GWs in Section 4.2."

L180-181- Figure A3 showing negative correlations, however Figure A2 displays positive correlations. What is the possible mechanisms behind the two opposite correlations at two height regions?

440 The effect of ozone recovery on GWs is indirect: changes in the stratospheric circulation imply changes in the filtering of the spectrum of upward propagating GWs, which via momentum deposition influence the wind.

L192-193- 'twice less' – does it mean half? Why the trends in meridional winds (MWs) are half that of zonal winds (ZWs) as no climatological means are presented here? Further how do the trends in ZWs and MWs are comparable to extract the

445 common trend structure?

Yes, we revised to "twice smaller". The trends in MWs are half that of ZWs because climatological means of MWs are also smaller compared to ZWs. This was not apparent from our figures. Therefore, we revised our figures accordingly.

L194- 'southward'... do you mean further poleward? Clarify.

450 We changed to "soutward directed circulation" in the manuscript.

L194-195- What could be the reason for negative trend at two different heights at two different stations?

The GW and GW trend in Fig. 3 is very different as well, and consequently the mean winds differ as well. As mentioned elsewhere, local background-wind shear and critical-level filtering vary with longitude and season, and that site-specific GW

455 sources can shift where wave drag changes project onto the mean flow.

L200-What are these 'constraints' – elaborate.

As also pointed out in our response to Ref. #1, we deleted the second part of this sentence ("...due to ...") to avoid confusion.

460

L202-203- I could not see any positive trends over Davis below 90 km in the summer from Figure 2.

We revised this paragraph to "Our analysis at Davis shows negative and positive trends, respectively, in meridional winds below 80 km in the austral summer, i.e. a descent of the amplitude for meridional winds. Similarly, Vincent et al. (2019) reported using MF radar wind measurements at Davis that meridional velocities peak at lower altitudes. These secular changes

465 in meridional circulation are linked with upwelling and may result in a shift of the summer mesopause due to adiabatic effects and vice versa (Smith et al., 2010; Ramesh et al., 2020)."

L204- What is the reason for increased upwelling? Figure 2 – What are A-F labels represent in the figure?

As mentioned above, we revised the paragraph. Further, the labels **A–F** are now explained in the caption of Fig. 2.

470

L207- I could not find the positive trends (statistically significant) of ZW at 80 km over any station in September.

We agree. We replaced "September" by "October".

L208- Does the figure A3 signifies the trends in ZW or simply ZW variation at 82 km for November?

[Figure]

**Figure R2.** Climatological sum of meteor counts in November at Davis.

It shows ZW variations at 82 km in November. We revised to "...(see time series in Fig. A3)".

L207-211 – How significant the ZW trends at 80 km estimated by MRs as the meteor counts are minimum at this height?

The uncertainty indicated in e.g. Fig. A3 by dashed lines is derived based on meteor counts at a given altitude and month. The uncertainty apparently does not limit the trend estimates at this altitude and month. We also enclose the climatological profile of number of meteors used for fitting (see Section 3.1) in November at Davis (see Fig. R2) to document the altitudinal dependence.

L209- How was the zonal momentum flux calculated and then the trends in the same?

We added an short paragraph introducing how the zonal momentum fluxes are calculated.

L211-215- I could see the eastward momentum flux is not statistically significant over Rothera especially after September below 90 km; however it is significant over Davis and what could be the reason despite the two stations being on the same latitude band.

While stations are located on the same latitude band, longitudinal differences in GW sources and background winds can change both the net MF and its trend as shown in Fig. 3. We also avoid over-interpreting site-to-site differences. This part was significantly changed based on the comments of both refs.

L217- The main concern is that no result was showed for the link/coupling between GWs and the ozone recovery. What is the direct impact of ozone (depletion/recovery)on the GWs as well as the MLT winds? Must be specified.

The effect of ozone recovery on GWs is indirect: changes in the stratospheric circulation imply changes in the filtering of the spectrum of upward propagating GWs.

This paragraph was heavily revised based on comments of Ref. #2.

L216-217 and Figures 2 & 3 – I could see the westward trend of GW momentum flux during May-Sept. over Davis from Fig. 3A, however, the trend in ZWs are eastward during this period above 80 km. Same for Rothera above 90 km. Why? Comparing to Figs. 2B and 3B, what could be the reason for negative and positive trends of ZWs below and above 90 km during Nov.-Feb. over RioGrande? Is there any link between GW MF and ZW trends? What is the 'slope' in Fig. 3B?

The situation in May-September should not be related to ozone recovery as the ozone hole just starts to build up in September. Therefore the mechanisms for the circulation changes are less clear. Still, wind filtering of GWs could be the reason for parts of the observed patterns. Westward (eastward) trends of the zonal wind in the stratosphere often coincide with opposite, i.e., eastward (westward) zonal wind trends at altitudes 80-90km. This can be explained by reduced westward (eastward) GW momentum fluxes due to changes in the stratospheric wind filtering of GWs, shifting the net momentum fluxes more into eastward (westward) direction at 80-90km, which is more favorable for the forcing of eastward (westward) winds at those altitudes. Explanation of trends at even higher altitudes may be complicated by the formation of secondary GWs when primary GWs break at altitudes 80-90km and contribute to the reversal of the summer mesospheric zonal wind jets. At Rio Grande secondary GWs should be even more dominant, making it generally difficult to explain directions and trends of GW momentum fluxes. See also our reply to Reviewer 1.

L219-222 – I could see transition of westward trends to eastward around at 90 km over all the stations during Nov.-Feb.

L219-222 describes the negative trend in summer (easterly) circulation in the SH, i.e. January and February, and thus a delayed transition to winter (westerly circulation in the SH.

L219-222- I wonder what is the transition refer here with Jan.-Feb. being the summer in SH? Further what are 'weaker' and 'stronger' refer to as the eastward MF is more significant over Davis than that over Rothera during Jan. – Feb.? Also why the westward MF above 90 km is more persistent all the year over Rothera than Davis being both the stations located at the same latitude band?

As mentioned above, we mean the transition between summer and winter in SH.

L223-229- Explain how these values are calculated and include appropriate references. Further is this valid over other two stations as no results are presented? Figure 5 – This figure needs more explanation with additional details to understand the importance of these comparisons.

We decided to move original Figs. 4,5,6 to the Appendix as they provide a methodological sensitivity test for one representative time series. They also underscore robustness of our trend estimates throughout the manuscript.

L230-236 – Since the period 2005-2021 is longer than a 11-year solar cycle, this time series is enough to derive the trends. If that is the case, what is the need of Fig. 6 and not enough details are explained to understand this figure and its relevance to Fig. 4. Further fewer years i.e. 8 years are inadequate to retrieve the trends.

As mentioned above, we decided to move these figure into the Appendix. We also clarify that TTE quantifies detectability given the observed variance and autocorrelation; it is not a statement that 8 years is universally sufficient, but that for this specific time series and slope, detection would be expected within ∼8 years under the assumed noise model.

L239-240 & L241-242- Which correct…above 200 km or 500 km? Further what are the GW parameters like vertical wavelength and periods as both ground based and space borne measurements have different horizontal and vertical resolutions and periods.

Both values are correct: 500km for radar, 200km for SABER. This means that there is some overlap of the SABER and radar observational filters, however, there are large spectral ranges that are seen by only SABER (longer than 500km) or by the radars (shorter than 200km; and close to the 200km limit radars should be much more sensitive to GWs than SABER).

SABER can observe vertical wavelengths longer than about 4km (twice the vertical field of view of the instrument). In terms of wave periods: limb viewing satellite instruments are sensitive to GWs of intrinsinc wave periods longer than about 1...2 hours (Alexander et al., 2010, their Fig. 8b).

L243-What is the data source and how was the GW potential energy (PE) calculated and trends in it? Should be explained. Why it was limited up to 90 km only?

We added "derived from SABER observations".

While SABER data are available even above 100 km, GW characteristics are derived only from 30 to 90 km (Ern et al., 2018) to avoid altitudes of increased measurement noise. We noted it in Section 2.3.2.

L243-245- Did you omit the time series of MR GW PE as same as shown in Fig. 7 for better comparisons?

We omit time series of GW PE derived from SABER because they include large gaps. Using this approach we try to avoid spurious trends as mentioned in the manuscript.

L245-246- I could see the negative trends instead positive.

We revised the sentence to "We document similarly slightly positive trends in March at Davis and Rothera, and in May at Rio Grande — indicative of stronger $E_{pot,V}$ in the strengthening..."

L247- What does weakening winds mean…eastward or westward?

[Figure]

**Figure R3.** Trend comparison (shading) of zonal monthly mean winds at the location Rothera for the period 2005–2021 omitting years 2016–2018. Solid (positive values) and dashed (negative values) contours represent wind climatology. Hatching \\\\ and //// shows where the p-values of the MK test are $< 0.05$ and $< 0.01$, respectively.

We revised to "weakening westward-directed winds in these months".

L249- Why does the negative trends stronger at Davis than at Rothera despite at the same latitude?

We speculate here. While stronger changes happen below 90 km at Davis as shown in Fig. 2., at Rothera we observe stronger changes above 90 km outside of the range of GW PE from SABER.

L251- What are net momentum fluxes? Explain.

A given volume over in given time interval often contains multiple GWs. The momentum flux vectors of these waves can have different directions. Averaging over these momentum flux vectors will result in "net" momentum flux (there can be cancellation effects).

We added "(averaging over momentum flux vectors)".

L255 - Epot and not Epot.

We introduced a proper LaTeX form here.

L261-262- Here and wherever applicable – Do MR observations have continuous measurements or any data gaps exist? No where this was mentioned. Further data gaps could be serious concern when compare with other datasets like WACCM.

As documented in Noble et al. (2024), the largest period of poor data quality takes place at Rothera for the period 2016–2018. Since the main advantage of our technique lies in the robustness of the fitting method that permits the decomposition of time series with data gaps, we detect missing raw data only between October 2017 and February 2018. We have not found any impact of these data gaps on our trend estimates when omitting years 2016–2018 (see Fig. R3). We added this statement in the manuscript.

[Figure]

**Figure R4.** Comparison of zonal monthly mean winds at the locations of Rio Grande, Rothera and Davis simulated by SD-WACCM-X at 96 km for July.

Figure 8 – It is well known that the WACCM doesn't reproduce the winter westerlies (eastward winds) over mid-high latitudes in the upper MLT. I am wondering how did the authors obtain positive trends above 85 km in the model. Further as trend means changes on a time scale longer than a solar cycle (11-year), the data shown here are inadequate. How were the trends in winds from the models calculated?

It is true that the WACCM doesn't reproduce the winter westerlies (eastward winds) over Rio Grande, Rothera and Davis. The positive trends above 85 km in the model are common at all locations (see Fig. R4) and also observed by meteor radars (see Fig. 2).

We agree the length of the common period 2008–2017 may not be adequate to assess trends due to a possible influence by the 11-year solar cycle. However, this period starts and ends during solar minima, respectively. Additionally, we reproduce similar trends for the longer period 2008–2021 in Fig. 2 as discussed in the manuscript.

Figures 8A & D were already shown and repeated from Figure 2. Same for Figures 9 and 10.

It is true but not for the common period 2008–2017 between both models and the MR station. We try to be consistent as much is possible.

Figure 8F – As the WACCM reproduces winter southward and northwards winds below and above 90 km, why the trends shown in this figure are insignificant and do not agree with the observations in the upper MLT?

We realized that there was a bug in the visualization script. While, according to new Fig. 8**F**, WACCM reproduces the strengthening of meridional winds during winter below 80 km, the trends are not statistically significant as in GAIA (see Fig. 8**E**). We modified our descriptions of figures accordingly.

L274- If the focus of this study was the mesosphere, why the trends shown for all layers from 20 km? This applies to all figures and lines.

The trends at lower altitudes in the stratosphere affect wave filtering of upward propagating GWs. These GWs contribute to the forcing of the circulation in the mesosphere, which means that the trends in the stratosphere are relevant for explaining trends in the mesosphere.

L275- Specify here what is meant by beginning and end of summer as I could not find negative trends in the beginning and positive trends in the end of the summer?

We reformulated the paragraph to: "Figs. 8**D**–**F** show that a negative trend in the meridional wind in JJA is only reproduced by GAIA, similarly to Rothera in Figs. 10**D**–**F**. This trend appears at all layers between 20 and 100 km, though it is statistically significant only at particular layers. The pattern of negative trend in Apr/May preceding the positive one in Aug/Sep at Davis in Figs. 9**D**–**F**, is not captured by SD-WACCM-X."

L279- Remove '.' I wonder why most of the trends are statistically insignificant? Further why the trends are positive in Nov./Dec. as it contradicts with the westward zonal flow below 90 km in winter including Oct. – Mar.?

We replaced dot with comma. While positive trends around 80 km in Nov/Dec are stat. significant at the station and in GAIA, SD-WACMM-X reveals significance in December which can be seen by the edge of Fig. 9. We suggested the explanation using Fig. 3 where the westward zonal momentum flux weakens at this altitude and in these months and consequently the westward zonal flow weakens as well.

L281-282- I could not see the WACCM reproduction of the long-term changes in zonal winds.

As mentioned above there was a bug in the visualization script. Now Fig. 9 shows that the WACCM reproduction of the long-term changes in zonal winds.

L283-285- I trust the focus of this study is of the mesosphere and not the below layers, and I could not see any negative and positive trends in the meridional winds during the said period at the heights of meteor radar observations.

As mentioned above there was a bug in the visualization script. The paragraph was reformulated and documents that trend slopes in SD-WACCM-X (Fig. 9**F**) indicate a similar pattern, but they differ above 75 km where the equatorward and poleward

meridional flow is weakened in Oct/Nov/Dec and in Jun/Jul, respectively.

L289- I wonder how the trends are positive if the zonal flow is westward during Nov. – Feb. below 90 km?

We replaced with "the weakening of easterlies".

L290- I could see the opposite trends during these months below 80 km.

Indeed the sentence should be as follows: "Positive and negative trends in the meridional wind in Apr/May/Jun and Aug/Sep/Oct, respectively, are well reproduced only by GAIA."

L291 – Again I still believe the focus of this study is mesosphere and not stratosphere.

Of course, the trends at lower altitudes in the stratosphere affect wave filtering of upward propagating GWs. These GWs contribute to the forcing of the circulation in the mesosphere, which means that the trends in the stratosphere are relevant for explaining trends in the mesosphere.

L296- I wonder no result has been presented on the ozone recovery in terms of concentration/vmr and this led to misinter-pretation on the wave dynamics and in turn the MLT winds over Antarctic.

As we argued above, ozone hole area is a standard metric how to document ozone recovery, also used in previous studies (e.g. Venkateswara Rao et al., 2015). We suggest changes in the filtering conditions for upward-propagating GWs as one of the possible mechanisms.

L296-297- Is this 'evident' from any result shown here?

We revised to "The recovery of the Antarctic ozone layer has led to a weakening and earlier breakdown of the stratospheric polar vortex, particularly evident in the zonal winds in the stratosphere (e.g. Banerjee et al., 2020)."

L301 – In this case, no results on PWs and tides were shown. It is very important to focus on these wave phenomena especially over the polar latitudes in the context of ozone variability.

We revised to "One mechanism for coupling between the stratosphere and the MLT region is wave coupling by PWs, GWs of various scales, and by thermal tides. In our work we provide additional evidence for the role of coupling by GWs using both MR-derived momentum flux and satellite-derived potential energy ($E_{pot,V}$)."

L305 – Which 'analysis'?

We replaced it with "study".

Figure A2 – What is the significance of this figure at 30 km? Why the blue lines (ozone hole area) are different from those in Figure A3?

We included p-values in both figures. Time series in Fig. A2 are longer because it shows correlation between the MERRA2 reanalysis and ozone hole area. We also added that that the time ranges between those figures differ.

670

**References**

Alexander, M. J., Geller, M., McLandress, C., Polavarapu, S., Preusse, P., Sassi, F., Sato, K., Eckermann, S., Ern, M., Hertzog, A., Kawatani, Y., Pulido, M., Shaw, T. A., Sigmond, M., Vincent, R., and Watanabe, S.: Recent developments in gravity-wave effects in climate models and the global distribution of gravity-wave momentum flux from observations and models, Quarterly Journal of the Royal Meteorological Society, 136, 1103–1124, https://doi.org/https://doi.org/10.1002/qj.637, https://rmets.onlinelibrary.wiley.com/doi/abs/10.1002/qj.637, 2010.

Banerjee, A., Fyfe, J. C., Polvani, L. M., Waugh, D., and Chang, K.-L.: A pause in Southern Hemisphere circulation trends due to the Montreal Protocol, Nature, 579, 544–548, https://doi.org/10.1038/s41586-020-2120-4, 2020.

Bremer, J., Schminder, R., Greisiger, K., Hoffmann, P., Kuerschner, D., and Singer, W.: Solar cycle dependence and long-term trends in the wind field of the mesosphere/lower thermosphere, J. Atmos. Sol.-Terr. Phys., 59, 497–509, https://doi.org/10.1016/S1364-6826(96)00032-6, 1997.

de Wit, R. J., Janches, D., Fritts, D. C., Stockwell, R. G., and Coy, L.: Unexpected climatological behavior of MLT gravity wave momentum flux in the lee of the Southern Andes hot spot, Geophysical Research Letters, 44, 1182–1191, https://doi.org/10.1002/2016GL072311, https://agupubs.onlinelibrary.wiley.com/doi/abs/10.1002/2016GL072311, 2017.

Ern, M., Trinh, Q. T., Preusse, P., Gille, J. C., Mlynczak, M. G., III, J. M. R., and Riese, M.: GRACILE: a comprehensive climatology of atmospheric gravity wave parameters based on satellite limb soundings, Earth System Science Data, 10, 857–892, https://doi.org/10.5194/essd-10-857-2018, https://www.earth-syst-sci-data.net/10/857/2018/, 2018.

Ern, M., Preusse, P., and Riese, M.: Intermittency of gravity wave potential energies and absolute momentum fluxes derived from infrared limb sounding satellite observations, Atmospheric Chemistry and Physics, 22, 15 093–15 133, https://doi.org/10.5194/acp-22-15093-2022, https://acp.copernicus.org/articles/22/15093/2022/, 2022.

Merzlyakov, E., Murphy, D., Vincent, R., and Portnyagin, Y.: Long-term tendencies in the MLT prevailing winds and tides over Antarctica as observed by radars at Molodezhnaya, Mawson and Davis, J. Atmos. Sol.-Terr. Phys., 71, 21–32, https://doi.org/https://doi.org/10.1016/j.jastp.2008.09.024, 2009.

Newman, P. A., Nash, E. R., Kawa, S. R., Montzka, S. A., and Schauffler, S. M.: When will the Antarctic ozone hole recover?, Geophysical Research Letters, 33, https://doi.org/https://doi.org/10.1029/2005GL025232, https://agupubs.onlinelibrary.wiley.com/doi/abs/10.1029/2005GL025232, 2006.

Noble, P. E., Hindley, N. P., Wright, C. J., Cullens, C., England, S., Pedatella, N., Mitchell, N. J., and Moffat-Griffin, T.: Interannual Variability of Winds in the Antarctic Mesosphere and Lower Thermosphere Over Rothera (67°S, 68°W) During 2005–2021 in Meteor Radar Observations and WACCM-X, Journal of Geophysical Research: Atmospheres, 129, e2023JD039 789, https://doi.org/https://doi.org/10.1029/2023JD039789, https://agupubs.onlinelibrary.wiley.com/doi/abs/10.1029/2023JD039789, e2023JD039789 2023JD039789, 2024.

Portnyagin, Y., Forbes, J., Fraser, G., Vincent, R., Avery, S., Lysenko, I., and Makarov, N.: Dynamics of the Antarctic and Arctic mesosphere and lower thermosphere regions—I. The prevailing wind, Journal of Atmospheric and Terrestrial Physics, 55, 827–841, https://doi.org/https://doi.org/10.1016/0021-9169(93)90024-S, https://www.sciencedirect.com/science/article/pii/002191699390024S, 1993.

Portnyagin, Y., Merzlyakov, E., Solovjova, T., Jacobi, C., Kürschner, D., Manson, A., and Meek, C.: Long-term trends and year-to-year variability of mid-latitude mesosphere/lower thermosphere winds, Journal of Atmospheric and Solar-Terrestrial Physics, 68, 1890–1901,

https://doi.org/https://doi.org/10.1016/j.jastp.2006.04.004, long-term Trends and Short-term Variability in the Upper, Middle and Lower Atmosphere, 2006.

710 Ramesh, K., Smith, A. K., Garcia, R. R., Marsh, D. R., Sridharan, S., and Kishore Kumar, K.: Long-Term Variability and Tendencies in Middle Atmosphere Temperature and Zonal Wind From WACCM6 Simulations During 1850–2014, Journal of Geophysical Research: Atmospheres, 125, 1–20, https://doi.org/10.1029/2020JD033579, 2020.

Smith, A.: Interactions Between the Lower, Middle and Upper Atmosphere, Space Science Reviews, 168, 1–21, https://doi.org/10.1007/s11214-011-9791-y, http://dx.doi.org/10.1007/s11214-011-9791-y, 2012.

715 Smith, A. K., Garcia, R. R., Marsh, D. R., Kinnison, D. E., and Richter, J. H.: Simulations of the response of mesospheric circulation and temperature to the Antarctic ozone hole, Geophysical Research Letters, 37, n/a–n/a, https://doi.org/10.1029/2010GL045255, http://doi.wiley.com/10.1029/2010GL045255, 2010.

Solomon, S., Ivy, D., Gupta, M., Bandoro, J., Santer, B., Fu, Q., Lin, P., Garcia, R. R., Kinnison, D., and Mills, M.: Mirrored changes in Antarctic ozone and stratospheric temperature in the late 20th versus early 21st centuries, Journal of Geophysical Research: At-

720 mospheres, 122, 8940–8950, https://doi.org/https://doi.org/10.1002/2017JD026719, https://agupubs.onlinelibrary.wiley.com/doi/abs/10.1002/2017JD026719, 2017.

Song, B.-G. and Song, I.-S.: Coupling of Long-Term Trends of Zonal Winds Between the Mesopause and Stratosphere in Southern Winter, Geophysical Research Letters, 51, e2023GL107014, https://doi.org/10.1029/2023GL107014, https://agupubs.onlinelibrary.wiley.com/doi/abs/10.1029/2023GL107014, 2024.

725 Strelnikova, I., Almowafy, M., Baumgarten, G., Baumgarten, K., Ern, M., Gerding, M., and Lübken, F.-J.: Seasonal Cycle of Gravity Wave Potential Energy Densities from Lidar and Satellite Observations at 54° and 69°N, J. Atmos. Sci., 78, 1359–1386, https://doi.org/10.1175/JAS-D-20-0247.1, 2021.

Venkateswara Rao, N., Espy, P. J., Hibbins, R. E., Fritts, D. C., and Kavanagh, A. J.: Observational evidence of the influence of Antarctic stratospheric ozone variability on middle atmosphere dynamics, Geophysical Research Letters, 42, 7853–7859,

730 https://doi.org/10.1002/2015GL065432, https://onlinelibrary.wiley.com/doi/abs/10.1002/2015GL065432, 2015.

Vincent, R. A., Kovalam, S., Murphy, D. J., Reid, I. M., and Younger, J. P.: Trends and Variability in Vertical Winds in the Southern Hemisphere Summer Polar Mesosphere and Lower Thermosphere, Journal of Geophysical Research: Atmospheres, 124, 11 070–11 085, https://doi.org/10.1029/2019JD030735, 2019.